# Motor neurons control blood vessel patterning in the developing spinal cord

Patricia Himmels[1,2], Isidora Paredes[1,2], Heike Adler[1,2], Andromachi Karakatsani[1,2], Robert Luck[1,2], Hugo H. Marti[3], Olga Ermakova[4], Eugen Rempel[4], Esther T. Stoeckli[5] & Carmen Ruiz de Almodóvar[1,2]

Formation of a precise vascular network within the central nervous system is of critical importance to assure delivery of oxygen and nutrients and for accurate functionality of neuronal networks. Vascularization of the spinal cord is a highly stereotypical process. However, the guidance cues controlling blood vessel patterning in this organ remain largely unknown. Here we describe a new neuro-vascular communication mechanism that controls vessel guidance in the developing spinal cord. We show that motor neuron columns remain avascular during a developmental time window, despite expressing high levels of the pro-angiogenic vascular endothelial growth factor (VEGF). We describe that motor neurons express the VEGF trapping receptor sFlt1 via a Neuropilin-1-dependent mechanism. Using a VEGF gain-of-function approach in mice and a motor neuron-specific sFlt1 loss-of-function approach in chicken, we show that motor neurons control blood vessel patterning by an autocrine mechanism that titrates motor neuron-derived VEGF via their own expression of sFlt1.

---

[1] Biochemistry Center, Heidelberg University, 69120 Heidelberg, Germany. [2] Interdisciplinary Center for Neurosciences, Heidelberg University, 69120 Heidelberg, Germany. [3] Institute of Physiology and Pathophysiology, Heidelberg University, 69120 Heidelberg, Germany. [4] Department of Stem Cell Biology, Centre for Organismal Studies, Heidelberg University, 69120 Heidelberg, Germany. [5] Institute of Molecular Life Sciences, University of Zurich, 8057 Zurich, Switzerland. Correspondence and requests for materials should be addressed to C.R.d.A. (email: carmen.ruizdealmodovar@bzh.uni-heidelberg.de).

Recent evidence demonstrates that neuro-vascular communication is crucial for the development of both the neuronal and the vascular network within the central nervous system (CNS)[1]. Blood vessels control neural stem cell differentiation[2,3], as well as migration of neuroblasts[4], of differentiated neurons[5] and of oligodendrocyte precursors[6]. Vice versa, neural cells modulate CNS vascularization by either expressing pro-angiogenic factors[7–9] or by acting as support for vessel formation and stabilization[10].

The vertebrate CNS is initially avascular and becomes vascularized by sprouting angiogenesis from a surrounding vascular plexus. In the developing spinal cord (SC) blood vessels sprout from the perineural vascular plexus (PNVP) and invade the SC at the ventral side[11,12]. Simultaneously, neuronal progenitor domains in the SC are being specified and organized in a ventral to dorsal pattern, and post-mitotic neurons are migrating towards their final location in the SC[13]. One of the best-characterized signals that controls CNS vascularization is vascular endothelial growth factor (VEGF)[8,12,14,15]. Neuroepithelium-derived VEGF controls the initial formation of the PNVP and the invasion of vascular sprouts into the neural tissue[8,12]. In addition, CNS vascularization is specifically controlled by other angiogenic signals, such as Wnt7 or GPR124 (refs 16,17). For proper SC vascularization blood vessels do not only need to sprout and grow but they also need to do it in a very precise manner, by invading the SC at specific locations and by following certain paths (stereotypical blood vessel patterning). Yet, the identity of the neural cells, the spatial cues and the signalling mechanisms that regulate this process remain largely unknown. It is also unclear whether patterning cues modulate VEGF signalling or act independently of the VEGF axis.

VEGF exerts its biological effects by interacting with two tyrosine kinase receptors, VEGF receptor-1 (VEGFR1, also known as fms-like tyrosine kinase, Flt1) and VEGF receptor-2 (VEGFR2, also known as fetal liver kinase, Flk1)[18]. Neuropilin-1 (NRP1), a receptor for class 3 Semaphorins, can also act as a VEGF receptor or co-receptor[19]. While VEGFR2 is considered as the main VEGF signalling receptor, Flt1 can either signal upon VEGF binding or act as a VEGF trap[20]. Post-transcriptional or post-translational modifications lead to a Flt1 isoform containing the transmembrane and the intracellular domain (mFlt1), or to a soluble isoform (sFlt1) lacking those two domains[21]. sFlt1 acts as a VEGF sink to titrate the amount of VEGF available for signalling[22,23]. In the endothelium, sFlt1, derived from the stalk cells of a vessel sprout, regulates the response of that particular sprout towards an external source of VEGF[23,24]. Whether a sFlt1-dependent mechanism exists at the neuro-vascular interface is unknown.

Gain-of-function studies in quail and chicken embryos showed that VEGF expression is required for proper blood vessel ingression into the SC[12,25]. Intriguingly, when blood vessel sprouts invade the SC from the ventral side they avoid the floor plate and the motor neurons (MNs), despite the fact that VEGF is expressed in those areas[26,27]. These previous findings raise the questions of what neuronal cell types are permissive for blood vessel sprouting from the PNVP? When do MN columns become vascularized in development? And how are VEGF expression and signalling regulated by the neuronal compartment to control the stereotypical blood vessel patterning in the SC? Here we have revisited those initial findings and answered these questions.

We describe a neuro-vascular communication mechanism by which MN columns prevent premature ingression of blood vessels into their location. In an autocrine mechanism MNs express VEGF to allow blood vessel growth, but at the same time express sFlt1 to titrate the availability of VEGF in order to pattern the vasculature and to block premature ingression of vessels into

MN columns during a developmental time window. This 'attraction' and 'repulsion' coming from the same cellular source (here MNs) proposes a novel mechanism that uses known angiogenic players to achieve proper tempo-spatial vascularization of the CNS.

## Results

**Blood vessels grow in a stereotypical pattern in the SC.** To understand the process of SC vascularization during mouse development in more detail, and to determine what neural domains control blood vessel sprouting from the PNVP, we analysed SC vascularization in correlation to different neuronal populations of the developing SC at brachial and thoracic levels. At E9.5 the PNVP (Isolectin-B4[+] (IB4[+]) (ref. 16)) is already formed but no blood vessels are invading the SC (Fig. 1a,b,f,j and Supplementary Fig. 1a,e,i,m,u). Invasion of blood vessels starts at around E10.0 and occurs ventrally, adjacent to the floor plate and lateral to MN columns. Subsequently, blood vessels, without invading the floor plate (Supplementary Fig. 1b–d), elongate through V3 post-mitotic interneurons (Sim1[+]) (Supplementary Fig. 1f–h) and in close contact with the p3 (Nkx2.2[+], neuronal progenitor domain generating V3 interneurons) and the pMN (Olig2[+], neuronal progenitor domain generating MNs) domains (Supplementary Fig. 1j–l,n–t). Starting at E10.5, ventral blood vessels surround MN columns (Isl1/2[+]) without invading them, while new blood vessels invade and branch in the dorsal SC (Fig. 1a,c,g) but avoid the most dorsolateral areas (Supplementary Fig. 1u–w). While MN columns remain still avascular at E11.5 (Fig. 1a,d,h), at E12.5 blood vessels can be detected (Fig. 1a,e,i). Similar vascularization pattern was also observed at other levels of the SC (Supplementary Fig. 1y-b'). Altogether, these results suggest that the ventral domains of floor plate, p3, V3 interneurons, pMN and MNs might control the initial steps of SC vascularization and patterning, with V3 interneurons, p3 and pMN neuronal domains acting as 'vascular attractant' sites, whereas floor plate and MNs acting as 'repellent' sites.

**VEGF is dynamically expressed during SC vascularization.** SC-derived VEGF is known to control CNS vascularization[8,12,25]. However, a precise characterization of the spatial and temporal VEGF expression, and the identity of the different neuronal populations expressing VEGF are lacking or fragmentary. Thus, we correlated the process of SC vascularization with VEGF expression and identified the cellular sources of VEGF. In situ hybridization (ISH) for Vegf combined with IB4 staining showed that Vegf expression changes dynamically during the process of SC vascularization. At E9.5 when the PNVP is formed, Vegf is uniformly expressed in the entire SC (Fig. 1j). At later developmental stages its expression in the ventral SC increases and becomes restricted to specific neuronal progenitor domains and groups of differentiated neurons, coinciding with ingression of blood vessel sprouts from the PNVP into the SC (E10.5), elongation and branching (E10.5-E12.5) (Fig. 1k–m). A combination of ISH with immunostaining for the MN marker Isl1/2 or for the p3 neuronal domain marker Nkx2.2 showed that at E10.5 and E11.5 Vegf is highly expressed in MN columns (Fig. 1o,p), in the floor plate (Fig. 1s,t) and in the neuronal progenitor zone (Fig. 1k–m,o–q,s–u). At E12.5, the time when MN columns become vascularized, Vegf expression is still present in MN columns (Fig. 1q). These results are consistent with previously described Vegf expression in the mouse and quail SC[8,12,25,27], and in addition identify the ventral neuronal populations acting as sources for Vegf.

Our correlative analysis of Vegf expression and SC vascularization (Fig. 1j–m) confirms that MN columns, expressing Vegf, are

not invaded by blood vessels despite the fact that VEGF is a potent vascular attractant (Fig. 1k,l). It suggests that during a developmental time window sprouting blood vessels are 'blind' to MN-derived VEGF and that perhaps VEGF signalling in these sprouting vessels is counteracted by other, yet unidentified, molecular mechanisms. These counteracting mechanisms should be released at later developmental stages (E12.5) when blood vessels enter MN columns still expressing VEGF.

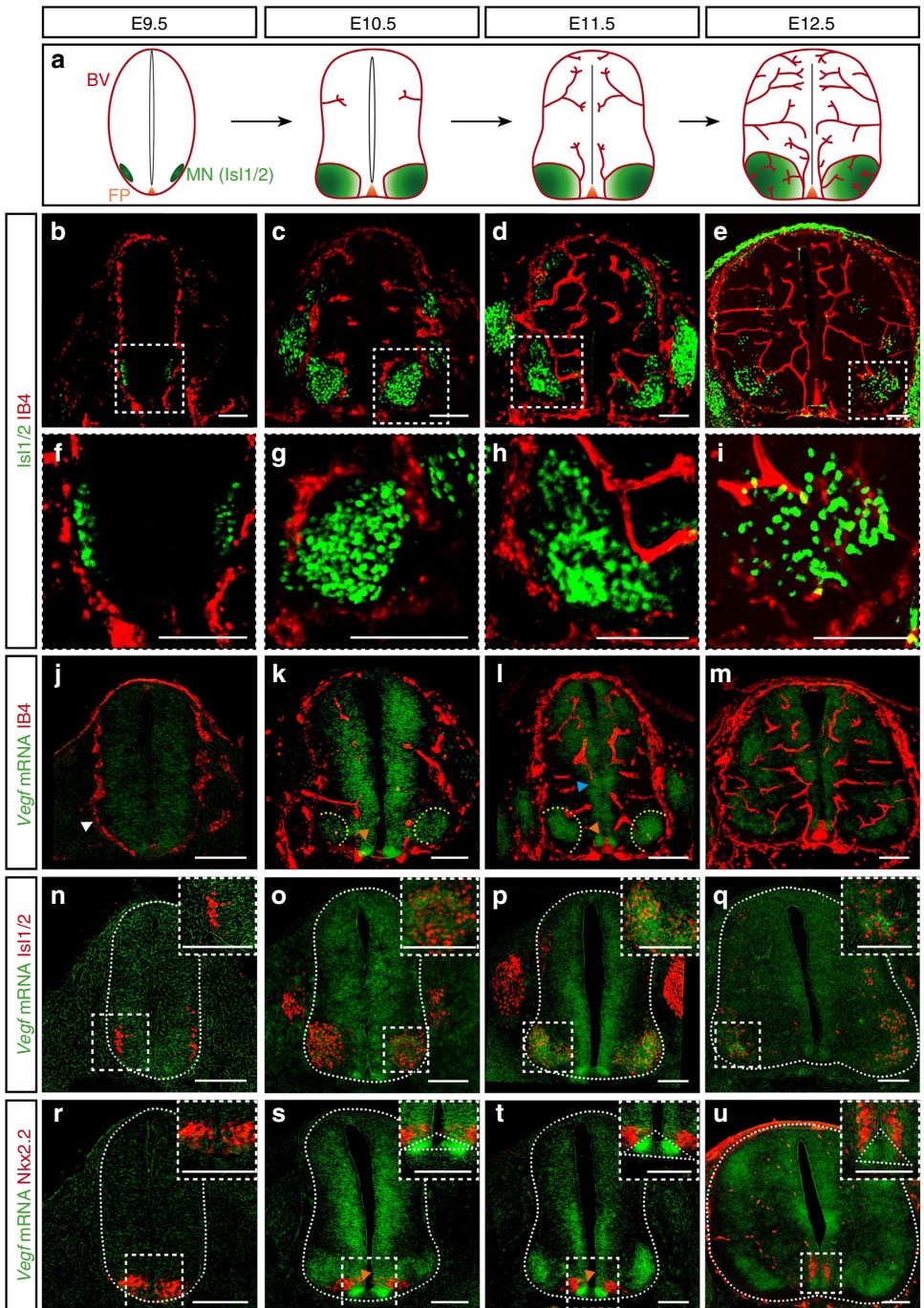

**Figure 1 | Blood vessel sprouts invade the developing mouse spinal cord at specific locations and grow in a stereotypical pattern.** (**a**) Scheme of SC vascularization during mouse development (E9.5 till E12.5), showing blood vessels (BV, red), the floor plate (FP, orange) and the MN columns (green). (**b–e**) Representative images of SCs at the developmental stages indicated, showing labelled endothelial cells (IB4$^+$) and post-mitotic MNs (Isl1/2$^+$). (**f–i**) Higher magnifications of insets in **b–e**. Note blood vessels stay outside the Isl1/2$^+$ domain till E12.5. (**j–m**) Representative images of blood vessel staining (IB4$^+$) in the SC combined with ISH for *Vegf* from E9.5 till E12.5 in mice. At E9.5 (**j**), *Vegf* is uniformly expressed in the entire SC (white arrowhead: PNVP). From E10.5 till E12.5 (**k–m**) *Vegf* expression becomes restricted to specific neuronal domains (yellow dotted lines: MN columns; orange arrowheads: FP; blue arrowhead: neuronal progenitors). (**n–q**) Representative images of ISH for *Vegf* combined with immunostaining for MNs (Isl1/2$^+$) confirming that *Vegf* expression is highly localized and increased in MN columns from E10.5 onwards. Insets show higher magnifications of MN columns. (**r–u**) Representative images of ISH for *Vegf* combined with immunostaining for the p3 neuronal progenitor domain (Nkx2.2$^+$) revealing expression of *Vegf* in the p3 domain and in the FP (orange arrowheads: FP (domain Nkx2.2$^-$, below Nkx2.2$^+$)). Insets show higher magnifications of the FP region (white dotted outlines: FP). Scale bars 100 μm.

**VEGF expression is regulated by HIF1α.** In an attempt to identify potential mechanisms that could regulate VEGF expression at the MN level we performed an *in silico* analysis. We first used *Cytoscape* (an open source software to analyse biomolecular network interactions[28]) to generate a list of potential VEGF regulators (Supplementary Data 1), which we then filtered by presenting it to an already available gene expression data set of SC MNs at E11.5 (ref. 29) (Supplementary Data 2). Enrichment analysis for signalling pathways of the obtained list of genes (using the *Enrichr* web-based software[30]) indicated that the response to hypoxia, and in particular HIF1α was highly enriched in MNs at E11.5 (Supplementary Fig. 2a,b). As HIF1α is a well-known regulator of VEGF expression[31] we investigated whether it regulates VEGF expression in SC MNs.

In fact, HIF1α expression and nuclear localization at MN columns correlated with VEGF expression during development. At E10.5 HIF1α was present in MNs and partly localized in the nucleus (Fig. 2a,d,g). While at E11.5 HIF1α was mainly localized in the nucleus of MNs (Fig. 2b,e,h), by E12.5 its expression pattern became partially excluded from the nucleus (Fig. 2c,f,i), consistent with higher oxygen levels due to the presence of blood vessels. Expression of carbonic anhydrase IX (CAIX), another HIF1α target gene, correlated with HIF1α expression in MN columns (Supplementary Fig. 2c–k). Moreover, mRNA expression of prototypical HIF target genes in microdissected MN columns correlated with HIF1α expression pattern (Supplementary Fig. 2l–n).

One of the major regulators of HIF1α function, and therefore of *Vegf* expression, is hypoxia[31]. Indeed, culture of isolated MN column explants from E11.5 mouse SCs in hypoxic conditions resulted in increased levels of *Vegf* as well as other HIF1α target genes (Fig. 2j–m). This increase of *Vegf* levels was HIF1α dependent as the addition of chetomin (a small molecule that attenuates HIF1α-dependent transcription[32]) abolished it (Fig. 2j–m).

Since SC vascularization and *Vegf* expression in chicken embryos are similar to mouse (Supplementary Fig. 3)[33], and HIF1α is expressed in MNs of chicken SCs[34], we used chicken embryos to confirm that *Vegf* expression is regulated by HIF1α *in vivo*. For this, we injected chetomin into the SC canal of chicken embryos. Analysis of *Vegf* expression revealed a strong decrease in *Vegf* levels in MN columns in the presence of the HIF1α inhibitor (Fig. 2n,o). The expression of *Vegf* in neuronal progenitors and floor plate was not completely abolished, indicating that the inhibition of HIF1α-dependent transcription was not completely blocked in these cells, or that *Vegf* expression is also regulated in a HIF1α-independent manner.

**VEGF overexpression results in premature MN vascularization.** If a balance between VEGF and inhibitory cues is needed to control the specific vascular patterning of the ventral SC, a deregulation of this balance towards higher concentrations of VEGF might result in premature invasion of blood vessels into the restricted areas. To specifically overexpress VEGF in the embryonic ventral SC at the time of SC vascularization, we used the transgenic mouse line (TgN(NSEVEGF)1651 (V1)), named from here on NSE-VEGF[tg]. These mice express the human VEGF$_{165}$ isoform under the promoter of the neuron-specific

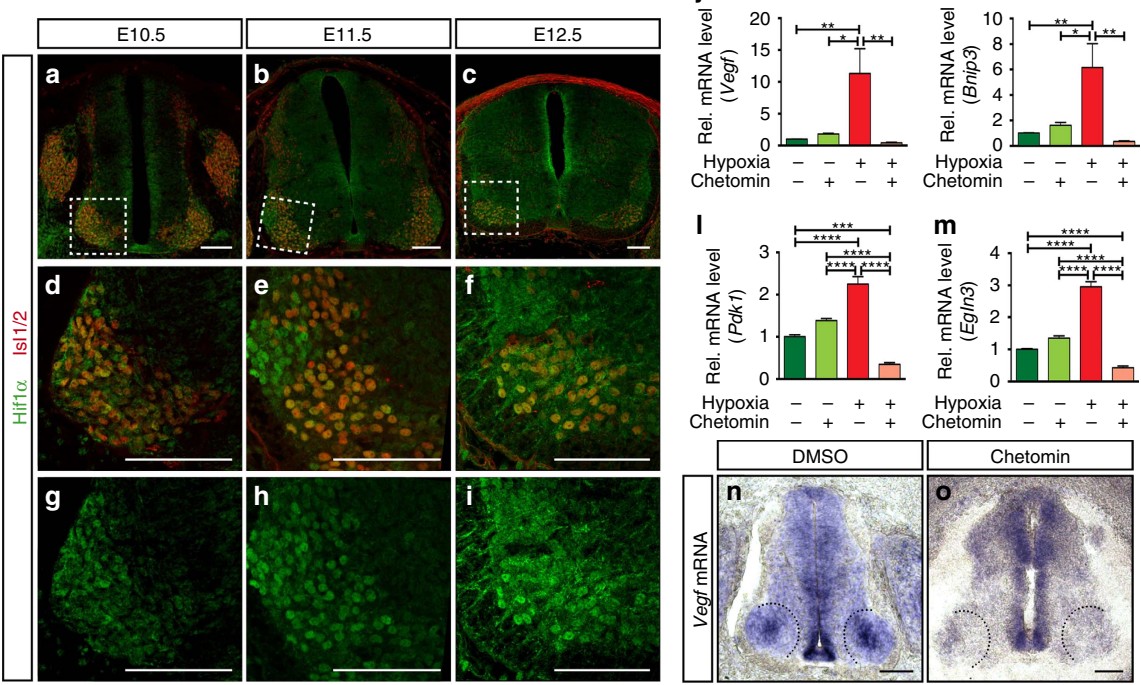

**Figure 2 | VEGF expression is regulated by HIF1α.** (**a–c**) Representative images of SCs at the developmental stages indicated, showing HIF1α staining in post-mitotic MNs (Isl1/2$^+$). (**d–i**) Higher magnifications of insets in **a-c** showing co-localization of HIF1α and Isl1/2$^+$ (**d–f**) or just HIF1α staining (**g-i**). Note the changes of HIF1α nuclear localization during development. (**j–m**) qRT-PCR analysis of changes in expression levels of *Vegf* (**j**) and other prototypical HIF target genes (*Bnip3* (**k**), *Pdk1* (**l**) and *Egln3* (**m**)), when explants from mouse E11.5 MN columns are cultured under normoxia (20% $O_2$) (green bars) or hypoxia conditions (1% $O_2$) (red bars), with or without chetomin. Data are represented as mean ± s.e.m. $n = 2$ individual experiments done in triplicates. (**j**) $*P = 0.0161$, $**P = 0.0086$ (normoxia versus hypoxia), $**P = 0.0082$ (hypoxia versus hypoxia + chetomin); (**k**) $*P = 0.0169$, $**P = 0.0056$ (normoxia versus hypoxia), $**P = 0.0018$ (hypoxia versus hypoxia + chetomin); (**l**) $***P = 0.0007$, $****P < 0.0001$; (**m**) $****P < 0.0001$. One-way ANOVA using Tukey's multiple comparisons test was performed for (**j–m**) and *P* Values were adjusted to account for multiple comparisons. (**n,o**) Representative images of ISH for *Vegf* mRNA in chicken embryos (HH27) treated either with control (DMSO) (**n**) or with chetomin (**o**). Note the absence of *Vegf* expression in MN columns of the chetomin-treated embryo (black dotted lines: MN columns). Scale bars 100 μm. ANOVA, analysis of variance.

enolase (NSE)[35] and its expression can be detected in SCs of E11.5 embryos (Supplementary Fig. 4a). Consistent with the *Nse* expression pattern (Supplementary Fig. 4b–e), $hVegf_{165}$ is much higher expressed in microdissected MN columns than in the dorsal SC of E11.5 NSE-VEGF[tg] embryos (Supplementary Fig. 4f). ISH for *Vegf* using a probe that detects both *mVegf* and *hVegf* also revealed that NSE-VEGF[tg] mice expressed higher levels of *Vegf* (*mVegf* + *hVegf*) in the ventral SC, in particular in MN columns, when compared with WT embryos (Supplementary Fig. 4g,h).

E9.5 WT and NSE-VEGF[tg] embryos showed no anatomical differences and no blood vessel sprouts inside the SC (Supplementary Fig. 5a–c). At E10.5 and E11.5, NSE-VEGF[tg] embryos displayed similar sizes and anatomical features as WT embryos (Supplementary Fig. 5c) but showed small haemorrhages in the brain and in the SC (Supplementary Fig. 5d–g). Quantification of blood vessel density in the entire SC revealed ~1.5-fold increase in NSE-VEGF[tg] embryos compared with WT embryos at E10.5 (Fig. 3a,c,i) and E11.5 (Fig. 3e,g,k). Furthermore, detailed analysis of MN columns also showed a strong increase in blood vessel density in NSE-VEGF[tg] embryos (Fig. 3b,d,f,h,j,l). Comparable vascularization defects were observed at brachial and thoracic levels (Supplementary Fig. 5h–p).

To analyse blood vessel guidance, we determined the ingression pattern of blood vessels into the SC, focusing in particular on MN columns, where $hVEGF_{165}$ is highly expressed. Angles of angiogenic sprouts ingressing into the ventral half (0–90°) were calculated as previously described[12] (Supplementary Fig. 5q). As expected, in WT embryos, blood vessel ingression into the SC occurred in a highly stereotypical pattern with very few sprouts entering through MN columns (between 20–55°, Supplementary Fig. 5q) (Fig. 3m–o). Remarkably, in NSE-VEGF[tg] embryos blood vessels ingressed significantly more often through the MN columns (Fig. 3m–o) while the floor plate remained avascular as in WT embryos (Supplementary Fig. 5r,s).

Finally, we questioned whether the observed premature ingression of blood vessels into MN columns would have any consequences in MN development. Analysis of total number of all MNs (Islet1/2[+]) or of lateral MNs (FoxP1[+], Islet1/2[+]) revealed no differences between WT and NSE-VEGF[tg] embryos (Supplementary Fig. 6a–f). However, analysis of the distribution of all MNs or of lateral MNs showed that MNs were positioned abnormally and intermixed in NSE-VEGF[tg] embryos (Supplementary Fig. 6g–t). In addition, evaluation of motor axons exiting the SC showed that while in WT embryos axons leave the SC in a fasciculated manner, in NSE-VEGF[tg] these were highly defasciculated (Supplementary Fig. 6u–x).

Collectively, these results indicate that an increased, thus unbalanced, expression of VEGF in the ventral SC is sufficient to deregulate ventral vascular patterning. Although we cannot exclude a direct effect of the overexpressed VEGF on MNs, these data indicate that in the presence of blood vessels MNs do not cluster properly and suggest that MN columns should remain avascular until E12.5 to assure their proper grouping and positioning within columns. Furthermore, they also support a model where inhibitory signals in MN columns can be overcome by VEGF overexpression.

**MN conditioned medium has a pro-angiogenic effect**. To investigate whether MN columns might express inhibitory factors that prevent blood vessel ingression, and whether these are long-range or short-range cues, we performed several *in vitro* approaches with isolated MN explants from E11.5 WT SCs (Supplementary Fig. 7a). To test whether MNs secrete inhibitory cues, we obtained conditioned medium (CM) from MN explants and tested its angiogenic potential on human brain microvascular endothelial cells (HBMECs)[36]. First, we analysed HBMEC migration using a transwell assay and found that in contrast to an inhibitory angiogenic effect, HBMECs migrated significantly more in the presence of MN-CM than in control conditions (Fig. 4a). Recombinant VEGF was used as a positive control (Supplementary Fig. 7b). Next, we performed a matrigel-based tube formation angiogenic assay[37]. Consistently, when HBMECs were treated with MN-CM, an increase in tube length and tube branching was observed (Fig. 4b,d,f,g). We then compared the effect of MN-CM with the CM from explants of the area dorsally adjacent to MN columns, which is permissive to blood vessels, and which we termed ingression site (immunostaining for MN or interneuron markers confirmed the nature of the explants (Supplementary Fig. 7c–f)). Quantification of tube length and branching showed that MN-CM has similar pro-angiogenic effects as the CM from ingression site explants (Supplementary Fig. 7g,i,k,l).

Since VEGF is present in MN-CM (Supplementary Fig. 7m) we tested whether the observed pro-angiogenic effect can be reversed by the addition of recombinant Flt1/Fc[22], which traps VEGF. Indeed, Flt1/Fc blocked the pro-angiogenic effects observed in the transwell and in the tube formation assay (Fig. 4a,c,e–g). Interestingly, Flt1/Fc could not block the pro-angiogenic effects of ingression site-CM (Supplementary Fig. 7b,h,j–l), suggesting that VEGF is not the sole pro-angiogenic factor in this SC region and that perhaps other factors (i.e., Wnt7a/b) might be responsible for the angiogenic effect.

These results indicate that MN columns secrete VEGF, which is pro-angiogenic. They further suggest that inhibition of blood vessel invasion into MN columns might be then mediated by a short-range, locally expressed and retained, inhibitory cue.

**MNs express short-range anti-angiogenic factors**. To explore the possibility that a short-range inhibitory cue is responsible for preventing blood vessels from invading MN columns, we developed two *neuro-angiogenic* assays (Supplementary Fig. 7a). First, we tested the probability of HBMEC tubes to approach a MN explant by co-culturing them on matrigel (Supplementary Fig. 7a). Time lapse imaging, as well as quantification of images taken at the end of the experiment, showed that while HBMECs formed tubes and remained attached to ingression site explants, in the presence of MN explants tubes formed at a distance from the explant and very few touched them (Fig. 4h–j, Supplementary Movie 1 and 2). Explant size was similar in all experimental conditions (Supplementary Fig. 7n).

Second, we tested the capacity of HBMECs to migrate towards a MN explant. For this, we modified the traditional scratch assay[37] by placing the explant in the scratch area. At 6 h after the initiation of the assay no difference was observed when comparing the migration of HBMECs alone or in the presence of ingression site or MN explants (Fig. 4n). After 20 h, HBMECs cultured alone or in the presence of ingression explants had closed or almost closed the gap, respectively (Fig. 4k,m,n). However, HBMECs in the presence of MN explants had only covered 88% of the gap by then (Fig. 4l,n). By 30 h, this effect was still observed (Fig. 4n). We also used MN explants from NSE-VEGF[tg] mouse embryos or their WT littermates (explant size was similar for both genotypes, Supplementary Fig. 7o). While no differences were observed at 6 h (Fig. 4q), at 20 h and 30 h HBMECs in the presence of NSE-VEGF[tg]-MN explants had closed the gap significantly more than HBMECs in the presence of WT-MN explants (Fig. 4o–q). This result further supports our hypothesis that VEGF overexpression is able to overcome the potential inhibitory cues present in MNs.

Together, these data suggest that MNs express a membrane-bound or a poorly diffusible factor, potentially associated with the extracellular matrix (ECM), that is capable of patterning the vasculature within a short radius of action. Thus, vessels would be allowed to grow around MN columns but would be prevented from invading them. Moreover, our results suggest that a tightly controlled balance between this inhibitory factor(s) and VEGF is required for proper blood vessel patterning.

**mFlt1 and sFlt1 are expressed in MN columns.** Since sFlt1 and mFlt1 have an effect on vascular patterning by acting as negative angiogenic regulators via VEGF trapping[20,22], and sFlt1 remains

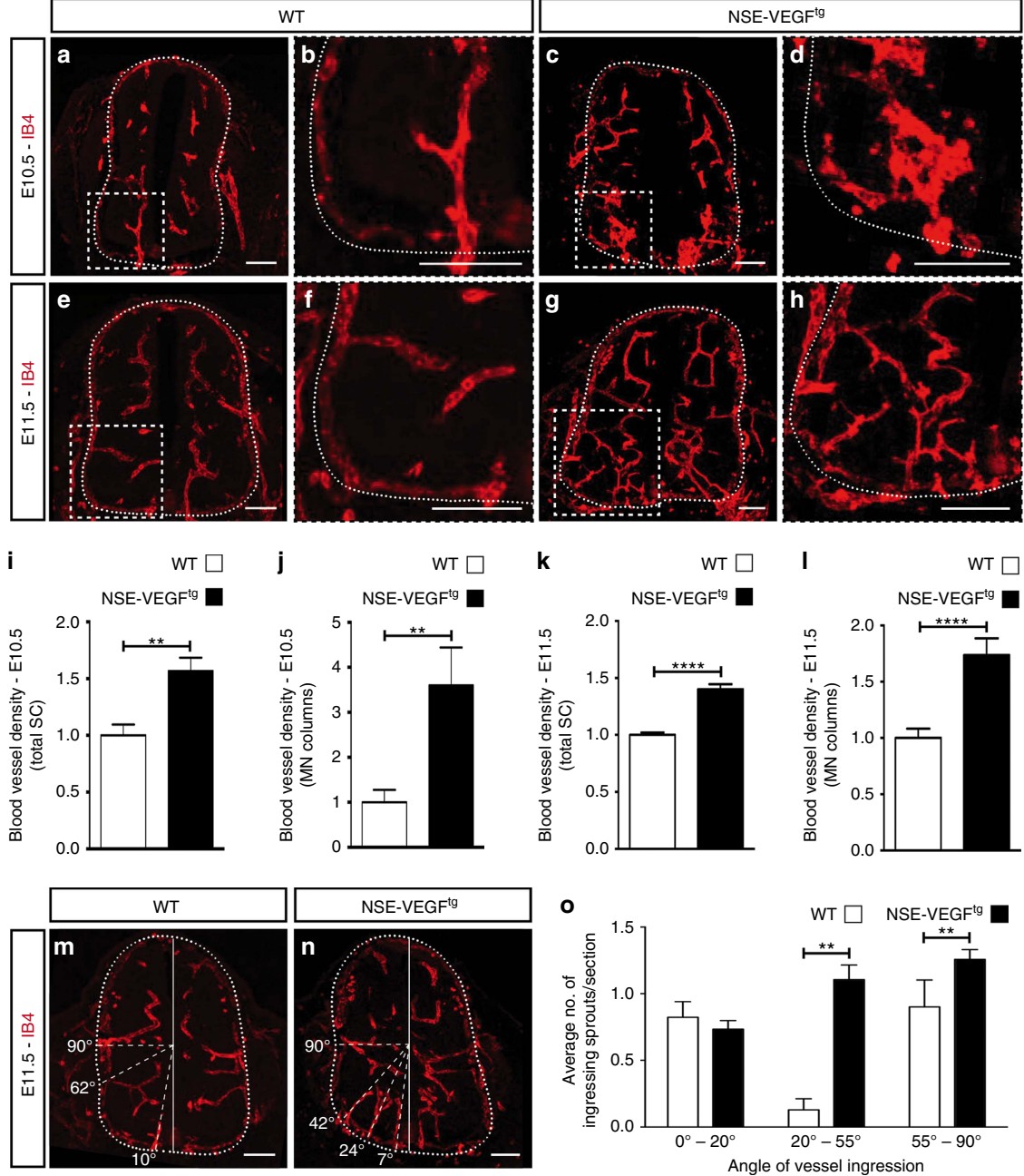

**Figure 3 | Overexpression of VEGF in the ventral spinal cord leads to premature blood vessel ingression into MN columns.** (a–h) Transverse SC sections stained for endothelial cells (IB4$^+$) at E10.5 (a–d) and at E11.5 (e–h) reveal increased blood vessel density in the SC of NSE-VEGF$^{tg}$ (c,d,g,h) compared with WT (a,b,e,f) embryos, especially at the level of MN columns (insets). Higher magnifications of insets in (a,c,e,g) are shown in (b,d,f,h), respectively. (i–l) Quantification of blood vessel density in the entire SC (i,k) or within MN columns (j,l) for E10.5 (i,j) and E11.5 (k,l) WT and NSE-VEGF$^{tg}$ embryos. For E10.5: $n = 5$ WT embryos and $n = 11$ NSE-VEGF$^{tg}$ embryos from three different litters. **$P = 0.0023$ (total SC), **$P = 0.0094$ (MN columns). For E11.5: $n = 15$ WT embryos and $n = 41$ NSE-VEGF$^{tg}$ embryos from eight different litters. ****$P < 0.0001$. (m,n) Representative images of blood vessel ingression analysis in the ventral half of the SC in a WT embryo (m) and its NSE-VEGF$^{tg}$ littermate (n). Dashed lines indicate the respective ingression angle for the different ingressing sprouts. (o) Quantitative analysis of blood vessel ingression angles into the ventral half (0°–90°) of E11.5 SCs. Note that in NSE-VEGF$^{tg}$ embryos blood vessels ingress significantly more often through the MN columns (20°–55°) compared with WT littermates. $n = 11$ WT embryos and $n = 24$ NSE-VEGF$^{tg}$ embryos from six different litters. **$P = 0.0038$ (20°–55°), **$P = 0.0080$ (55°–90°). Data are represented as mean ± s.e.m. Scale bars 100 μm.

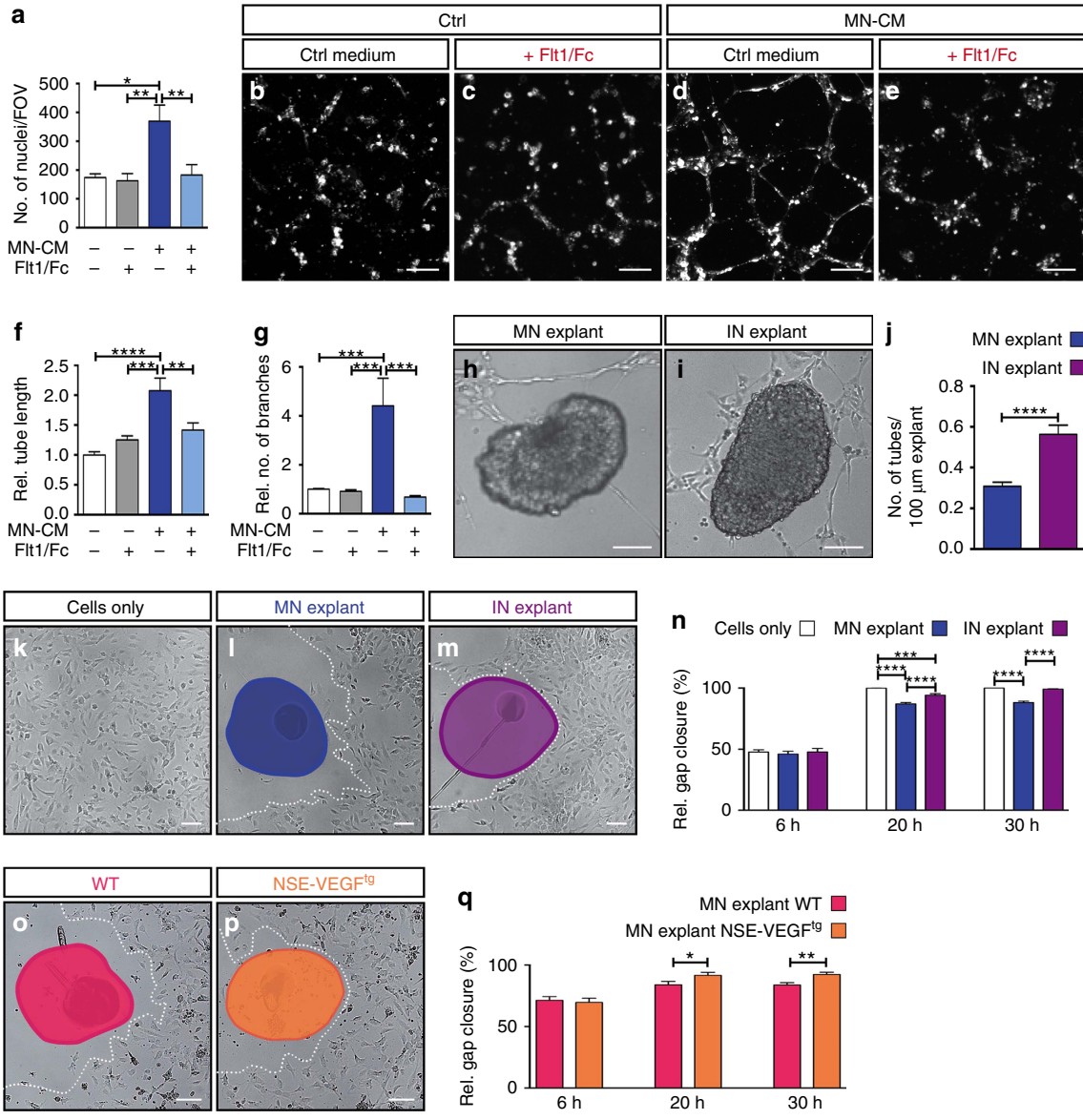

**Figure 4 | MN explants regulate vascular patterning by a defined balance of pro- and anti-angiogenic factors.** (**a**) Quantification of transwell assay with HBMECs in the presence of control (ctrl) or MN-CM, and with or without $1 \mu g\, ml^{-1}$ Flt1/Fc. FOV: field of view. $n = 3$ individual experiments (in triplicates). \*$P = 0.0104$, \*\*$P = 0.0016$ (Flt1/Fc versus MN-CM), \*\*$P = 0.0053$ (MN-CM versus MN-CM + Flt1/Fc). (**b**–**e**) Representative images of tubular structures formed by HBMECs in ctrl conditions (**b,c**) or in the presence of MN-CM (**d,e**), with (**c,e**) or without (**b,d**) $1 \mu g\, ml^{-1}$ Flt1/Fc. (**f**) Quantification of tube-length. \*\*$P = 0.0057$, \*\*\*$P = 0.0003$, \*\*\*\*$P < 0.0001$. (**g**) Quantification of number of branches. \*\*\*$P = 0.0002$ (MN-CM versus MN-CM + Flt1/Fc), \*\*\*$P = 0.0005$ (ctrl versus MN-CM), \*\*\*$P = 0.0007$ (Flt1/Fc versus MN-CM). For (**f**) and (**g**) $n = 4$ individual experiments done in triplicates. (**h,i**) Representative images of 'tube-touching'-assay showing HBMEC tubes and their tendency to 'touch' MN explants (**h**) or ingression site (IN) explants (**i**). (**j**) Quantification of the number of tubes touching either MN explants or IN explants, assessed per $100 \mu m$ explant perimeter. $n = 45$ MN explants and $n = 26$ IN explants from three independent experiments. \*\*\*\*$P < 0.0001$. (**k**–**m**) Representative images of the scratch area around the explant after 20 h of HBMECs only (**k**), HBMECs together with MN explant (**l**) or HBMECs together with IN explant (**m**). (**n**) Quantification of the relative gap closure after 6, 20 and 30 h. $n = 19$ explants per condition from at least three independent experiments. \*\*\*$P = 0.0004$, \*\*\*\*$P < 0.0001$. (**o,p**) Representative images of scratch assay after 20 h of HBMECs together with MN explant from WT (**o**) or MN explant from NSE-VEGF[tg] embryos (**p**). (**l,m,o,p**) White dotted lines: cell front. (**q**) Quantification of the relative gap closure after 6 h, 20 h and 30 h. $n = 12$ explants per condition, three independent experiments. \*$P = 0.0259$, \*\*$P = 0.0043$. Differences in gap closure between WT in (**n**) and (**q**) are probably due to different genetic background of embryos used. One-way ANOVA using Tukey's multiple comparisons test was performed for (**a,f,g,n,q**). $P$ Values were adjusted to account for multiple comparisons. Data are represented as mean ± s.e.m. Scale bars $100 \mu m$. ANOVA, analysis of variance.

matrix-associated[38,39], we hypothesized that they could be putative candidates for counteracting VEGF derived from MN columns and thus for controlling SC vascularization patterning.

Using an anti-Flt1 antibody that recognizes the extracellular domain, and thus detects mFlt1 and sFlt1 (mFlt1 + sFlt1 is from here on termed Flt1)[40], we found Flt1 in blood vessels

(Fig. 5a,b,e,f,i,j), radial glia (Fig. 5c,d) and highly expressed in MN columns, both at E10.5 and at E11.5 (Fig. 5a–h). However, by E12.5 Flt1 expression was reduced in MN columns and became more homogeneous within the SC (Fig. 5i–l). We next determined the mRNA expression pattern of each isoform using RNAscope probes designed to detect either *mFlt1* or *sFlt1*

specifically. Consistent with our immunohistochemistry data, we detected expression of both isoforms in blood vessels and in MNs, which was markedly decreased in MNs at E12.5 (Fig. 5m–p; Supplementary Fig. 8a,b).

Having confirmed the expression of sFlt1 in MNs, we next determined whether it is indeed secreted by MNs but associated to the ECM. ELISA analysis revealed that treatment of MN explants with heparin treatment (known to induce the release of

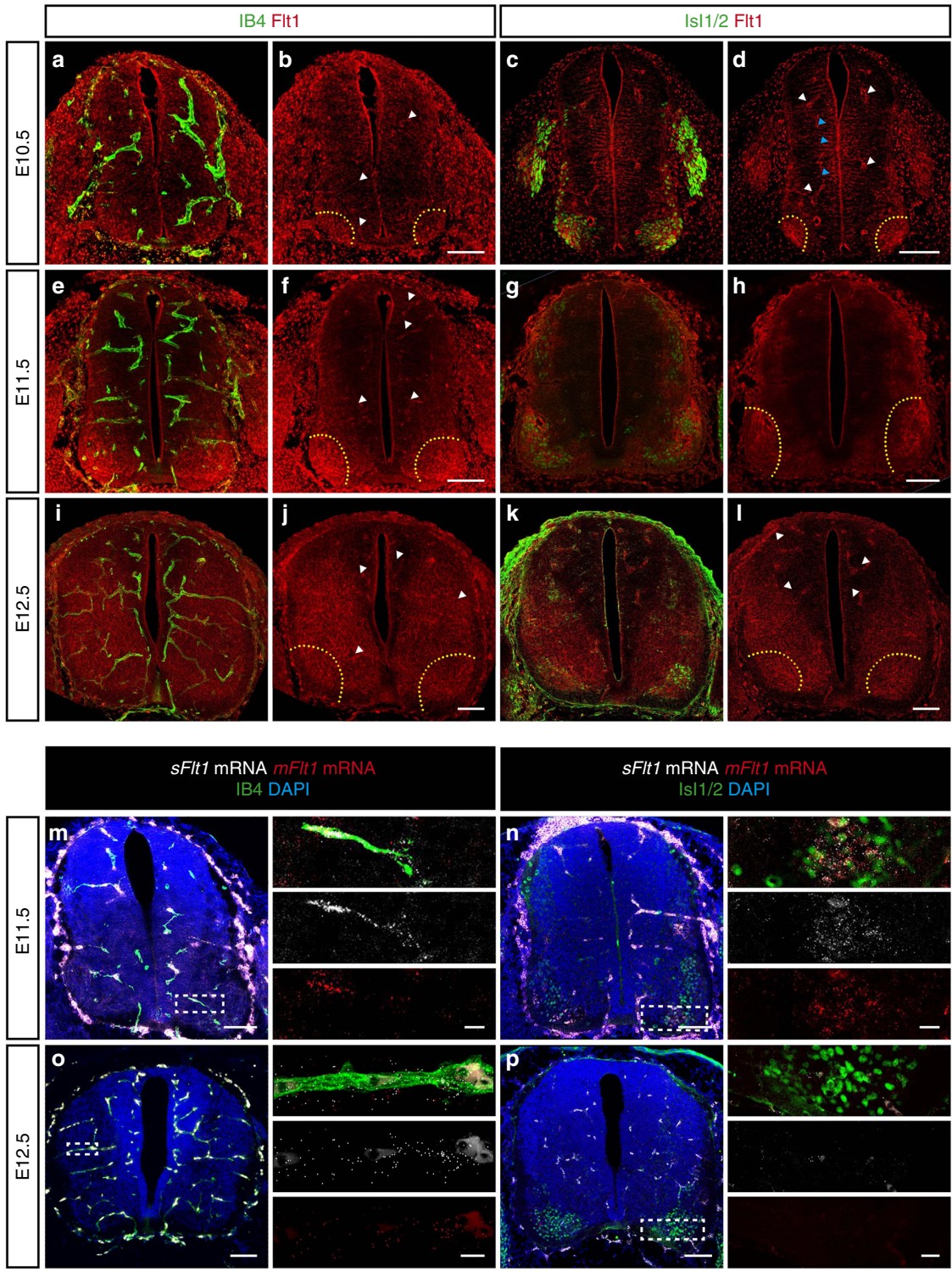

**Figure 5 | mFlt1 and sFlt1 are expressed in MN columns.** (**a,b,e,f,i,j**) Representative images of immunostaining for Flt1 (mFlt1 + sFlt1) combined with blood vessel staining (IB4$^+$) at E10.5 (**a,b**), E11.5 (**e,f**) and E12.5 (**i,j**). (**c,d,g,h,k,l**). Representative images of immunostaining for Flt1 (mFlt1 + sFlt1) and for Isl1/2$^+$ (post-mitotic MNs) at E10.5 (**c,d**), E11.5 (**g,h**) and E12.5 (**k,l**). Note Flt1 can be detected in blood vessels (white arrowheads), radial glia (blue arrowheads, identified by its typical morphology), as well as in MN columns (yellow dotted outlines). (**m–p**) Representative images of RNAscope Multiplex Fluorescent Assay using specific probes to detect *mFlt1* and *sFlt1* combined with staining for nuclei (DAPI$^+$) and either staining for blood vessels (IB4$^+$) (**m,o**) or immunostaining for MNs (Isl1/2$^+$) (**n,p**) at E11.5 (**m,n**) and E12.5 (**o,p**). Scale bars 100 μm. Right panels: higher magnifications of insets. Scale bars of insets 10 μm.

ECM-retained sFlt1[41]) significantly increased the amount of sFlt1 present in MN-CM (Supplementary Fig. 8c). Consistent with the release to the medium, quantification of Flt1 (mFlt1 + sFlt1) in the corresponding lysates of MN explants revealed a significant reduction of Flt1 in heparin-treated explants (Supplementary Fig. 8d).

These results support a model in which mFlt1 and/or sFlt1 can locally counteract VEGF-signalling by trapping VEGF, and thus regulate SC vascularization. Moreover, at E12.5 Flt1 expression becomes lower in MN columns. This change in expression pattern correlates with the invasion of blood vessel sprouts into MN columns at E12.5 and suggest that the VEGF-trapping action might be released by then.

**MN-specific loss of sFlt1 leads to premature vascularization.** To determine whether mFlt1 and/or sFlt1 prevent premature vascularization of MN columns, we applied *in ovo* RNAi electroporation in a cell-type specific manner. We used the MN-specific Hb9 promoter (Hb9p2.5distal/M-250-TATA[42,43]) to drive the expression of artificial microRNAs (miRNA) in MNs. The targeting vector also encoded EGFP to identify the electroporated MNs in the chicken SC (Supplementary Fig. 9a,b). Chicken SC vascularization starts at developmental stages HH17/19. By HH26/27 blood vessels surround MN columns and by HH30 blood vessels start invading them (Supplementary Fig. 3). Therefore, we unilaterally electroporated the SC at HH17 and analysed SC vascularization at HH29, using the non-electroporated side as internal control. In all conditions around 55% of Isl1/2$^+$ were targeted.

In control conditions (Hb9-EGFP-miLuc) chicken embryos displayed a normal vascular phenotype (Supplementary Fig. 9c–e). In contrast, when Flt1 (mFlt1 + sFlt1) was knockdowned using an artificial miRNA for Flt1 (Hb9-EGFP-miFlt1#1) (Fig. 6b and Supplementary Fig. 9f,g), blood vessel density in MN columns of the electroporated side was 2.5-fold increased compared with the non-electroporated (control) side (Fig. 6a,c). Analysis of blood vessel patterning upon Flt1 knockdown showed more vessel sprouts ingressing through MN columns as in the control side (20°–80° angles, Fig. 6d). No difference in vessel patterning was observed in other regions of the ventral SC (Supplementary Fig. 9h). To determine whether sFlt1 was specifically required for preventing ingression of blood vessels into MN columns, we used an artificial miRNA targeting just sFlt1 (Fig. 6f; Supplementary Fig. 9i). The sole knockdown of sFlt1 resulted in an increase in blood vessel density in the electroporated MN column compared with the control non-electroporated side (Fig. 6e,g). Blood vessel patterning was also affected as vessel sprouts ingressed significantly more often through the MN columns (Fig. 6h), but not through other regions of the SC (Supplementary Fig. 9j). Similar results were obtained with a second set of artificial miRNA-based vectors targeting a different coding sequence of either Flt1 or sFlt1 (Supplementary Fig. 9k–n). We then performed a rescue experiment by co-electroporating an Hb9-driven plasmid encoding mouse sFlt1 together with the Hb9-EGFP-misFlt1#1. Analysis of blood vessel density showed that the premature blood vessel ingression caused by Hb9-EGFP-misFlt1#1 was rescued (Fig. 6i–l), demonstrating the specificity of the miRNA used.

As we observed that the expression of Flt1 in MNs decreased at developmental stages when blood vessels start invading MN columns (Fig. 5), we asked whether maintained expression of sFlt1 in MNs would prevent blood vessels from entering these areas. For this, we electroporated the Hb9-sFlt1-HA plasmid in the embryonic chicken SC and analysed MN vascularization at HH30, when blood vessels have already naturally invaded MN columns (Supplementary Fig. 3f,l,r,x). Quantification showed that while in the non-electroporated side blood vessels were already

present in MN columns, in the sFlt1 overexpressing MN columns still very few vessels were detected (Fig. 6m–o).

Collectively, these experiments demonstrate that the pro-angiogenic effect of MN-derived VEGF is counteracted by the inhibitory effect of sFlt1 expressed by MNs. They show that sFlt1 levels need to be tightly regulated, as a partial reduction of sFlt1 is already sufficient to cause miss-guidance of blood vessel sprouts and premature MN column vascularization. Moreover, maintained expression of sFlt1 in MNs prevents blood vessel ingression at developmental stages when they normally start invading MN columns.

**Neuropilin-1 regulates sFlt1 expression in MNs.** To understand how sFlt1 is regulated in MNs, we screened for potential regulators of Flt1 expression or function in MNs using the same *in silico* approach as the one described above (Supplementary Data 3 and 4). Analysis of the generated list of genes (Supplementary Data 4) for enriched pathways revealed as top hits many of the known molecules involved in Flt1 functional regulation and signalling (Supplementary Fig. 10a,b).

We selected Neuropilin-1 (NRP1) for further analysis as we observed that its expression pattern in MN columns correlates with Flt1 expression. Indeed, *Nrp1* is highly expressed in MNs from E9.5 till E11.5 and decreases at E12.5 (Fig. 7a–e). Also, its expression pattern changes from being uniformly expressed within MN columns at E9.5-E10.5 to become restricted to specific MN subpopulations at E11.5-E12.5 (Fig. 7a–d). A similar expression pattern was observed in chicken embryos (Fig. 7f–k).

To explore whether NRP1 expression had any role on sFlt1 regulation in MNs, we silenced NRP1 in chicken embryo MNs using two different Hb9-driven artificial miRNA constructs (Fig. 7l; Supplementary Fig. 10c). Downregulation of *Nrp1* in MNs led to decreased *mFlt1* and *sFlt1* mRNA levels, without changing *Vegf* expression levels (Fig. 7l; Supplementary Fig. 10c), indicating that NRP1 transcriptionally regulates Flt1, and thus shifts the balance towards higher VEGF availability. The same vascular phenotype observed upon decreased sFlt1 levels was also observed upon NRP1 downregulation (Fig. 7m,n; Supplementary Fig. 10d). Thus, to determine whether this vascular phenotype was due to the decreased sFlt1 levels, we co-electroporated the NRP1 miRNA containing plasmid together with the Hb9-mouse sFlt1-HA plasmid. Co-expression of mouse sFlt1 rescued the phenotype induced by NRP1 knockdown (Fig. 7o,p), demonstrating that MNs control vascular patterning by a mechanism that depends on the transcriptional regulation of sFlt1 by NRP1.

**Discussion**

In the developing SC a variety of different processes has to be orchestrated in a precise temporally controlled manner. At the same time as neurons in the SC migrate and form networks, blood vessels invade the SC. To guarantee proper adaptation and synchrony with the neuronal environment, blood vessel formation, invasion and guidance within the SC needs to be highly controlled. In this study we report a novel mechanism by which MNs control blood vessel patterning in the developing SC in a tempo-spatial regulated manner (Fig. 8).

After the PNVP is formed, blood vessels sprout from the PNVP and invade the SC[7,12,16]. Once inside, blood vessels also pattern in a stereotypical manner. Due to the different binding affinities to the extracellular matrix, VEGF isoforms act as long-range and short-range vessel patterning cues and locally guide vessel sprouts in the CNS[44]. However, a single unifying model of blood vessel patterning solely controlled by VEGF cannot explain SC vascularization. In fact, our correlated analysis of *Vegf* expression with the patterning of blood vessels in the SC revealed that,

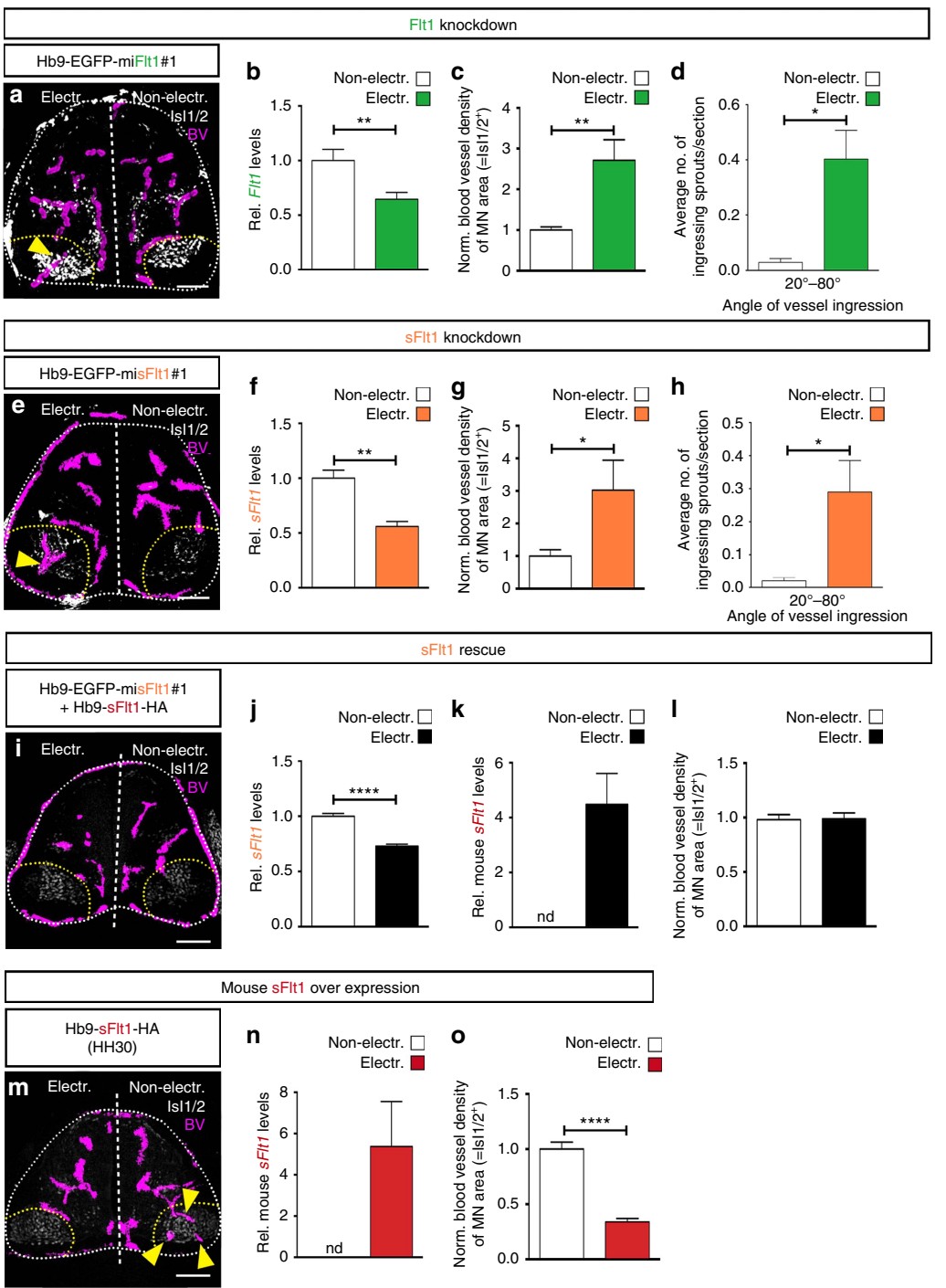

**Figure 6 | MN-specific deletion of sFlt1 leads to premature ingression of blood vessels into MN columns.** (a,e) Representative images of Hb9-EGFP-miFlt1#1 (a) or Hb9-EGFP-misFlt1#1 (e) electroporated chicken embryos showing blood vessel ingression (blood vessels (BV) labelled by ISH for *Vegfr2* are traced in purple) into MN columns in the electroporated side of the SC (yellow arrowheads) but not in the non-electroporated one. (b,f) qRT-PCR analysis confirms downregulation of *Flt1* (*mFlt1*+*sFlt1*) (b) or *sFlt1* (f) levels. $n = 4$, **$P = 0.0022$ for (b) and $n = 4$, **$P = 0.0042$ for (f). (c,g) Quantification of blood vessel density in MN columns (Isl1/2$^+$) shown as ratio between electroporated side and non-electroporated one. $n = 8$, **$P = 0.0012$ for (c) and $n = 8$, *$P = 0.0366$ for (g). (d,h) Quantification of the angle of ingression showing a significant increase in blood vessel sprouts ingressing into MN columns in the electroporated side but not in the non-electroporated one. $n = 11$, *$P = 0.0147$ for (d) and $n = 12$, *$P = 0.0408$ for (h). (i) Representative image of sFlt1 rescue experiment showing no blood vessel ingression into MN columns in the electroporated side of the SC as well as in the non-electroporated one. (j) qRT-PCR analysis confirms downregulation of chicken *sFlt1* levels in MN columns in the electroporated side. $n = 6$, ****$P < 0.0001$. (k) qRT-PCR analysis confirms expression of mouse *sFlt1* in MN columns in the electroporated side. nd = not detectable. $n = 6$. (l) Quantification of blood vessel density in MN columns (Isl1/2$^+$) shown as ratio between electroporated and non-electroporated side. $n = 4$, $P = $ ns. (m) Representative image of chicken SC at HH30, in which a mouse sFlt1-HA plasmid was unilaterally electroporated at HH17. (n) qRT-PCR analysis confirms expression of mouse *sFlt1* in MN columns of the electroporated side but it is not detectable (nd) in the non-electroporated side. $n = 7$. (o) Quantification of blood vessel density in MN columns (Isl1/2$^+$) shown as ratio between electroporated and non-electroporated side. $n = 9$, ****$P < 0.0001$. Scale bars 100 µm.

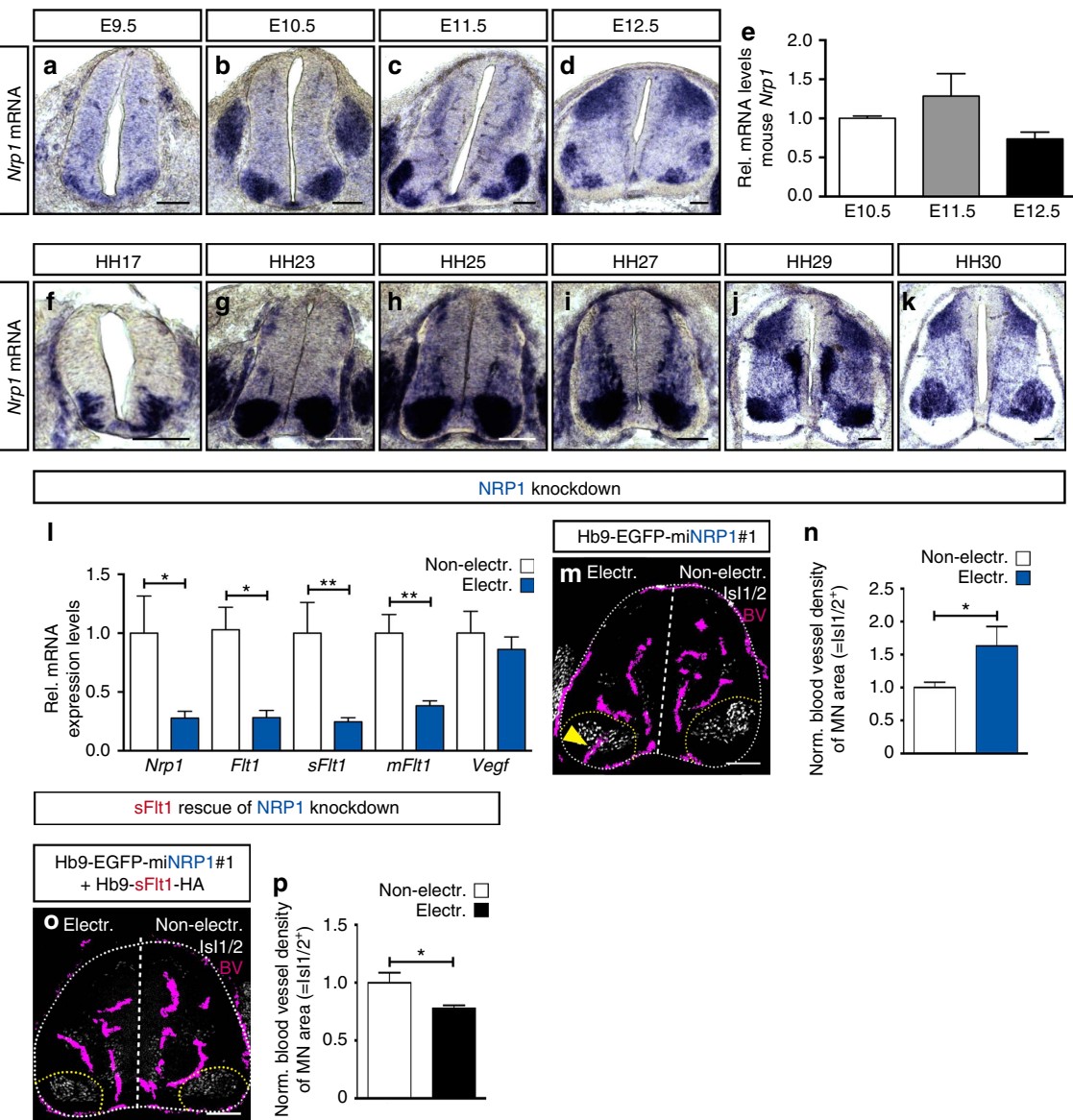

**Figure 7 | NRP1 regulates sFlt1 expression and as a consequence blood vessel guidance.** (**a–d**) Representative images of ISH for *Nrp1* in mouse SC sections at the developmental stages indicated. Note the changes in *Nrp1* expression levels and in its expression pattern during development. (**e**) qRT-PCR analysis of *Nrp1* in microdissected MN columns at E10.5, E11.5 and E12.5. Data are represented as mean ± s.e.m. $n = 4$ individual experiments done in triplicates. (**f–k**) Representative images of ISH for *Nrp1* in chicken SC sections at the developmental stages indicated. (**l**) qRT-PCR analysis showing downregulation of *Nrp1*, *Flt1* (*mFlt1 + sFlt1*), *sFlt1* and *mFlt1* levels in MN columns of chicken embryos electroporated with Hb9-EGFP-miNRP1#1. *Vegf* mRNA levels are unaffected. $n = 7$, *$P = 0.0437$ (*Nrp1*), *$P = 0.0306$ (*Flt1*), **$P = 0.0081$ (*sFlt1*), **$P = 0.0055$ (*mFlt1*) and $P =$ ns (*Vegf*). (**m**) Representative image of Hb9-EGFP-miNRP1#1 electroporated chicken embryos showing blood vessel ingression into MN columns in the electroporated side of the SC (yellow arrowhead) but not in the non-electroporated one. (**n**) Quantification of blood vessel density in MN columns (Isl1/2⁺) shown as ratio between electroporated and non-electroporated side in Hb9-EGFP-miNRP1#1 embryos. $n = 8$, *$P = 0.0274$. (**o**) Representative image of rescue experiment, in which Hb9-EGFP-miNRP1#1 is co-electroporated with a mouse sFlt1-HA plasmid. (**p**) Quantification of blood vessel density in MN columns (Isl1/2⁺) shown as ratio between electroporated and non-electroporated side. $n = 6$, *$P = 0.0349$. Scale bars 100 μm.

despite the high expression levels of *Vegf* in MN columns at E10.5-E11.5, blood vessels do not invade those regions until E12.5 (or until HH30 in chicken). To explain this highly regulated and stereotypical blood vessel patterning, here we favour the model where negative vascular cues also participate in SC vascularization. Indeed, our data showed that MN columns express both, long-range pro-angiogenic factors (secreted) as well as short-range inhibitory cues, which together balance the blood vessel response (Fig. 8).

Proper development of the vasculature requires a tightly controlled regulation of VEGF levels, as just a slight reduction or increase of VEGF results in severe vascular defects[14,45]. NSE-VEGFᵗᵍ embryos, which overexpress hVEGF₁₆₅ in MNs, have altered SC vascularization with premature blood vessel invasion of MN columns (Fig. 8), suggesting that VEGF availability around MN columns needs to be tightly regulated and that potential local counteracting/inhibitory cues should ensure this effect. Here, we identified sFlt1 as one of these inhibitory cues. Indeed, sFlt1 is expressed and locally retained at MNs, and its MN-specific knockdown leads to premature ingression of blood vessels into MN columns (Fig. 8). At the developmental stage when blood vessels normally invade MN

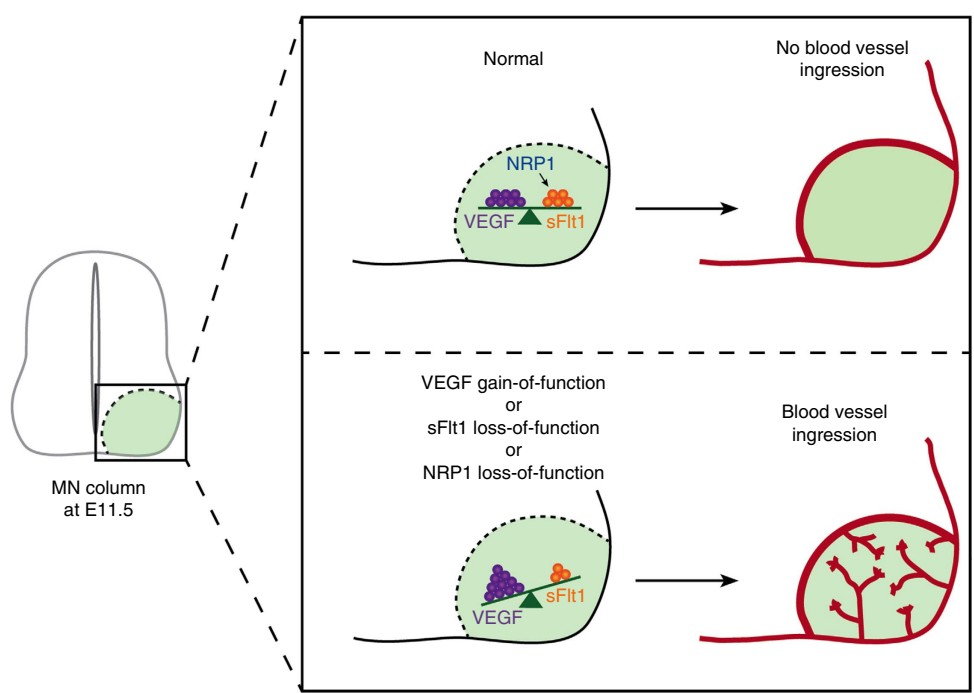

**Figure 8 | Model of how MNs control blood vessel patterning in the developing SC.** MNs express VEGF to allow blood vessel growth, but at the same time express sFlt1 to titrate the availability of their own produced VEGF in order to pattern the vasculature and prevent premature ingression of vessels into MN columns during a developmental time window. sFlt1 expression is transcriptionally regulated by NRP1. A deregulation of the balance towards higher concentrations of VEGF (VEGF gain-of-function, sFlt1 loss-of-function, or NRP1 loss-of-function (thus reduction of sFlt1)) results in the premature invasion of blood vessels into MN columns, indicating that VEGF levels need to be tightly titrated.

columns sFlt1 expression is lower than at earlier stages, suggesting that a reduction of sFlt1 releases the brake and allows vessel ingression by freeing higher amounts of VEGF. The tight regulation of VEGF availability in MNs is further demonstrated by the fact that a partial reduction of sFlt1 is sufficient to allow blood vessel invasion in MN columns.

Different molecular pathways are known to regulate Flt1 expression and function[24,46]. Here we show that NRP1 expressed in MNs regulates Flt1 (mFlt1 and sFlt1) at a transcriptional level and that knockdown of NRP1 results in premature ingression of blood vessels into MN columns (Fig. 8), which can be rescued by re-expression of sFlt1. During development NRP1 is required for proper positioning of MN somata within the SC[47], motor axon growth, guidance and fasciculation[48–51]. For these functions, NRP1 signalling in MNs is activated by Sema3A[48,49,51]. Our study shows a new role for MN-derived NRP1 as an indirect regulator of blood vessel patterning (Fig. 8) and positions NRP1 as a coordinator of neuro- and vascular patterning from the MN side. Future studies will elucidate the ligand that activates NRP1 in MNs to control sFlt1 expression.

Several studies have implicated endothelial-derived sFlt1 as a patterning factor that inactivates VEGF in vascular regions very close (from cell to cell) to the vessel sprout[23,24,52]. Expression of sFlt1 in the cornea also assures cornea avascularity[22]. Here we show that sFlt1 is expressed in both, blood vessels and MNs. Although we cannot exclude the possibility that endothelial-derived sFlt1 is also regulating vessel patterning along MN columns, the MN-specific deletion of sFlt1 indicated that its presence in MNs is crucial for proper SC vascularization. Therefore, a model where vessel- and MN-derived sFlt1 controls VEGF availability at both sides of the growing vessel is very likely to exist. Interestingly, sFlt1 is not the only VEGF receptor described to titrate VEGF levels in the CNS. In the developing retina, retinal neurons express VEGFR2 that traps VEGF via endocytosis and hence controls the direction of newly

formed vessel sprouts[53]. Thus, different neurons use different VEGF receptor-mediated mechanisms to titrate VEGF.

Our findings raise two questions. First, why do blood vessels remain outside MN columns for a specific developmental window? During E9.5 till E12.5 differentiated MNs are migrating into MN columns, MN cell bodies are clustering into columns, are establishing synaptic contacts and are extending their axons out of the SC to innervate specific muscles[54]. Thus, the presence of vascular sprouts could interfere in these processes. We speculate that MNs developed a mechanism to prevent that this happens and to allow vascularization when those processes are assured. In support of this, in NSE-VEGF[tg] embryos, where blood vessels invade MN columns prematurely, a MN clustering and axon fasciculation defect was observed. And second, why would a system express a ligand (VEGF) that then needs to be inactivated (via sFlt1)? We propose that sFlt1 traps the required amount of VEGF within MN columns in order to block premature blood vessel ingression, but that still certain levels of VEGF remain available, and are required to control endothelial cell survival, proliferation and growth of vessels surrounding the columns.

In summary, we show that VEGF signalling needs to be tightly regulated in order to guarantee proper blood vessel patterning in the developing SC. We demonstrate that MNs guide blood vessels by reusing classical angiogenic molecules in an alternative way. They present an autocrine mechanism by which they produce VEGF but at the same time themselves regulate VEGF availability by their own expression of sFlt1. Our work provides new insights into the regulatory mechanisms that control CNS vascularization, which could be relevant in understanding and designing new treatment strategies for different CNS pathologies.

## Methods
**Animals.** All experimental protocols, handling use and care of mice were approved by the local authorities and animal welfare officers and in accordance with the NIH 'Guide for the Care and Use of Laboratory Animals'. The TgN(NSEVEGF)1651

(V1) mouse line was previously described[35,55]. Wild-type littermates (WT) were used as controls. WT CD1 and C57Bl/6 mice were purchased from Janvier Labs and Charles Rivers.

All experiments involving chicken embryos were carried out according to the guidelines of the Cantonal Veterinary Office Zurich (Switzerland) as well as in accordance to local authorities and animal welfare officers of Baden-Wuerttemberg (Germany).

**Generation of miRNA constructs.** Constructs encoding miRNAs were generated as described[43]. Briefly, the mouse Hoxa1 enhancer III and minimal TATA box promoter from Hox-EGFP SDM in pBSK+ vector was replaced by Hb9p2.5distal/M-250-TATA box promoter. As negative controls, miRNA against firefly Luciferase (Hb9-EGFP-miLuc) was used. See Supplementary Table 1 for a complete list of miRNAs used in this study.

**Generation of Hb9-sFlt1-HA plasmid.** peak12-Hb9p25distal/M-250-TATA-EGFP was digested with HindIII and NotI. The mouse sFlt1 insert was amplified from mouse cDNA using specific primers with 3xHA-extension. The PCR products were digested with HindIII and NotI. Vector and insert were ligated following the manufactures instructions (NEB T4 DNA Ligase). The resulting plasmid was named Hb9-sFlt1-HA. See Supplementary Table 2 for primer sequences used.

**In ovo RNAi electroporation and blood vessel analysis.** In ovo RNAi was performed as described[56]. Briefly, chicken embryos (HH17) were injected with miRNA (800 ng μl$^{-1}$) into the central canal of the SC and electroporated subsequently (five pulses of 50 ms duration at 18 V). Eggs were placed back into a 38.5 °C incubator until HH29 or HH30 when SC vascularization was analysed. A control β-actin-RFP construct was co-electroporated as a positive control to assure that the electroporation worked[43]. For rescue experiments miRNA constructs (800 ng μl$^{-1}$) were co-electroporated with Hb9-sFlt1-HA (500 ng μl$^{-1}$). Mouse sFlt1-HA was not sensitive to misFlt1#1 (specific for chicken). Specificity was quantified by quantitative real-time PCR (qRT-PCR) analysis. For experiments, SC vascularization after electroporation was analysed by ISH for Vegfr2 (expressed in endothelial cells in chicken[25]) combined with immunostaining for the MN marker Isl1/2.

**In ovo injection of Hif1-α inhibitor.** All stock solutions were prepared in DMSO. Either control (DMSO-solvent) or chetomin (C9623, Sigma-Aldrich) (1 mg kg$^{-1}$) was injected into the SC canal of chicken embryos at stage HH24, and injections were repeated every 8 h till embryo dissection and subsequent analysis at HH27.

**RNAscope multiplex fluorescent assay.** mFlt1 and sFlt1 RNAscope probes were designed to detect either mFlt1 (C1 Mm-Flt-O1 443931) or sFlt1 (C2 Mm-Flt1-O2-C2 443941) in mice. RNAscope Multiplex Fluorescent Assay on fixed frozen tissue sections was performed according to manufacturer's instructions (Advanced Cell Diagnostics). Subsequently, blood vessels and MNs were visualized using Isolectin GS-IB$_4$ Alexa Fluor 568 conjugate (1:250, Invitrogen) for 2 h at room temperature (RT) or Isl1/2 (1:200, DSHB) 4 °C overnight, respectively. As negative control a RNAscope probe detecting dapb (bacterial gene) was used, and as positive control we used the 3-Plex-positive control RNAscope probe, targeting Polr2a, Ppib and Ubc, both provided by the manufacturer (Advanced Cell Diagnostics). Images were acquired with a Zeiss LSM 880 with Airyscan with 20x/0,8 Plan-APOCHROMAT. Image processing was performed using Zen 2.3 and the NIH ImageJ software.

**Tissue processing.** Mouse embryos were collected at different embryonic stages (E9.5 to E12.5) and fixed by immersion in 4% paraformaldehyde/PBS (diethyl pyrocarbonate (DEPC) treated) at 4 °C overnight. For RNAscope Multiplex Fluorescent Assay, mouse embryos were fixed by immersion in 4% paraformaldehyde/PBS (DEPC treated) at RT for 1 h. Chicken embryos were killed at different developmental stages (HH17-HH30)[57] and fixed by immersion in 4% paraformaldehyde/PBS (DEPC treated) at RT for 1 h. Afterwards, embryos were transferred to 25–30% sucrose/PBS-DEPC at 4 °C overnight and subsequently embedded in optimal cutting temperature (OCT) compound and stored at −80 °C. Serial 16-, 20- or 40-μm-thick transverse sections were cut using a cryostat (MICROM HM560) and collected on SuperFrost Plus slides (Menzel-Glaeser, Braunschweig, Germany).

**Whole mount staining on vibratome sections.** Whole mount vibratome immunostainings were performed as previously described with minor modifications[58]. Briefly, E11.5 embryos were fixed by immersion in 4% paraformaldehyde (PFA) at 4 °C for 6 h, washed with PBS, and embedded in 5.5% agarose, and sectioned at 500 μm on a vibratome (Leica VT1200S). Afterwards, the sections were blocked overnight at 4 °C in PBS-T (PBS with 1% Triton X-100) with 10% normal donkey serum. Following blocking, the sections were incubated with primary antibodies diluted in PBS-T with 1% normal donkey serum for 36 h. Sections were then washed five times for 30 min with PBS-T, and incubated with secondary antibodies diluted in blocking solution overnight at 4 °C. Finally,

sections were washed five times for 30 min at RT with PBS-T and the next day rinsed with PBS and mounted. Images were acquired on a Nikon AR1 confocal microscope with 20x/0,75 Plan-Apo λ.

**qRT-PCR.** RNA from tissue was isolated using the RNeasy Mini Kit (Ref:74104). Subsequently, DNAseI-treated (EN0521, Thermo Scientific) RNA was reverse-transcribed into cDNA using either Maxima Reverse Transcriptase (EP0742, Thermo Scientific) or SuperScriptVilo (11754-050, Thermo Scientific). mRNA expression levels were assessed by qRT-PCR using Fast SYBR Green Master Mix (00408995, Thermo Scientific), relative to the expression level of Gapdh or 18S (for mouse and chicken samples, respectively). For studying mRNA expression in normoxia and hypoxia conditions, mRNA levels were expressed relative to the housekeeping gene Rpl13a, due to its stable gene expression under these conditions[59]. The qRT-PCR primers used are indicated in Supplementary Table 3.

**Isolation of MN columns for qRT-PCR analysis.** Explants containing MN columns from E11.5 WT SC were microdissected via open book preparation (Supplementary Fig. 7a), followed by a dissociation step using a P-1000 pipette. Subsequently, the neural component was enriched by removing CD31$^+$ cells using CD31 MicroBeads (130-097-418, MACS Milteny Biotec) in combination with MS Columns (130-042-201, MACS Milteny Biotec) following the manufacturer's protocol for 'Isolation of endothelial cells from mouse neonatal brain—Section 2.2.2 (Enrichment of CD31$^+$ cells)' (Milteny Biotec)'.

**Histology and immunohistochemistry.** For immunohistochemistry the following primary antibodies were used at the indicated dilutions: Isl1 (40.2D6, 1:100, DSHB), Isl1/2 (39.4D5, 1:200, DSHB), Nkx2.2 (74.5A5, 1:100, DSHB), Olig2 (AB9610, 1:100, Millipore), Pax7 (1:10, DSHB), anti-h/m/r Hif-1a (AF1935, 1:100, R&D Systems), pAb anti-Carbonic Anhydrase IX/CA9 (NB100-417, 1:100, Novus Biologicals), TER-119 (MAB1125, 1:100, R&D Systems), rabbit anti-FoxP1 (ab16645, 1:1.000, Abcam), mouse anti-neurofilament-M (RMO 270, 1:1.500, ThermoFischer), anti-mouse Flt1 (103-M31, 1:100, ReliaTech GmbH), En-1 (4G11, 1:50, DSHB). The secondary antibodies that were used were donkey anti-mouse Alexa488 (715-545-150, 1:400, Jackson ImmunoResearch), goat anti-mouse Alexa488 (115-545-146, 1:400, Jackson ImmunoResearch), goat anti-rat Alexa568 (A11077, 1:400, Invitrogen), donkey anti-rabbit Alexa647 (711-605-152, 1:400, Jackson ImmunoResearch). Blood vessels were visualized using Isolectin GS-IB$_4$ Alexa Fluor 568 conjugate (I21412, 1:250, Invitrogen). Images were collected on a confocal microscope (Zeiss LSM 510 unit mounted on an Axiovert 200 M inverted microscope) with 10x/0,3 EC Plan-NEOFLUAR Objective and/or with 20x/0,8 Plan-APOCHROMAT and on a Nikon AR1 confocal microscope with 40x/1,3 Plan-Fluor Objective. Image processing was performed using Zen 2011 and the NIH ImageJ software.

**In situ hybridization.** Localization of mRNA was performed by ISH as follows: cryosections were hybridized with digoxigenin (DIG)-labelled antisense riboprobes (sequence information is provided in Supplementary Table 4). Hybridizations were carried out overnight at 68 °C. Riboprobes (except Shh probes) were detected by an alkaline phosphatase-coupled anti-digoxigenin antibody (diluted 1:500 or 1:5,000, Roche Diagnostics, Mannheim, Germany) and the compound nitroblue tetrazolium/5-bromo-4-chloro-3-indolyl phosphate (NBT/BCIP, Promega) was used as a chromogenic substrate for the alkaline phosphatase reaction. Exposure times ranged between 1 h and 2 days. No specific signal was obtained with sense probes in adjacent representative sections. Localization of Shh-mRNA was performed by fluorescent ISH (FISH) detection. Hybridization was carried out as mentioned above. Riboprobes were detected by an anti-digoxigenin-POD, Fab fragments antibody (diluted 1:500, Roche Diagnostics) and visualized with TSA Plus Fluorescein Fluorescence System (Perkin Elmer NEL741). Sections were observed under a fluorescence microscope Zeiss Axiovert 200 equipped with an AxioCam MRc camera.

**Quantification of blood vessel density and ingression angles.** The analysis of SC vascularization was always performed at the brachial and thoracic level, taking as a reference the forelimb of the mouse embryo. Analysis of blood vessel patterning was done at these levels as the vascularization pattern is similar along the SC, and brachial and thoracic regions are well established and accepted for studying neuronal progenitors, as well as their differentiation and migration in the developing SC[13,60].

The percentage of the area covered by blood vessels to the total area of the SC, or to the MN column area (=Isl1/2$^+$ cells) was quantified using the NIH ImageJ software. Values were normalized to the respective control (either control littermates (mouse) or MN column from the non-electroporated SC side (chicken)). Blood vessel ingression pattern was analysed quantitatively as previously described[12]. Briefly, the floor plate was taken as a reference and given the value of 0°. From there, the angles of ingressing sprouts were measured using the Angle tool of NIH ImageJ software. Angles were pooled in arcs of 5°. Angles, where the core of MN columns are located (20°–55° in mouse and 20°–80° in chicken), were determined by measuring the ventral-most as well as dorsal-most

point of Isl1$^+$ or Isl1/2$^+$ area in mouse WT or chicken embryos, respectively. For each experiment at least five separate transverse SC sections were analysed per animal. Quantification was done blind to the experimental condition. For better visualization of blood vessels in chicken embryos, using ImageJ the Segmentation Plugin 'Simple Neurite Tracer' was used and traced vessels were automatically filled out.

**MN quantification.** The number of total MNs was determined in 16 μm cryosections of brachial SC at E11.5 by the expression of determined transcription factors. Total number of MNs was obtained for at least three embryos per genotype by counting Isl1/2$^+$ and Isl1/2$^+$ FoxP1$^+$ cells per section in at least eight sections per embryo. The number of lateral motor columns was determined by co-expression of Isl1/2 and FoxP1 in ventro-laterally located cells. The FoxP1 lateral motor columns mean value was expressed as the percentage of the total number of MNs (Isl1/2$^+$ and Isl1/2$^+$ FoxP1$^+$).

**Image analysis for MN position.** We developed a method for determining MN soma position based on previously published procedures[61,62]. Coordinates from each Isl1/2$^+$ or FoxP1$^+$/Isl1/2$^+$ MN soma were obtained by processing binary 8-bit images using ImageJ software. A Cartesian coordinate system was established with the zero located at the ventro-lateral left border of the SC, with the $y$ axis parallel to the ventricle (ventral-dorsal (V-D) axis) and the $x$ axis at the MN ventral border (lateral-medial (L-M) axis). For reconstructing MN position of WT or NSE-VEGF$^{tg}$ embryos, data sets were plotted in scatter and contour density graphs using the bivariative histogram function 'hist3' of MATLAB. This function bins (divides the range of values into a series of intervals) the loaded data into a 10-by-10 grid. The $x$ and $y$ axes were imported as a column vector into MATLAB. After the data was divided into intervals, a two-dimensional projected view of intensities was generated. To do so, the data was interpreted as linear spaced vectors (with the 'linspace' function). This consists of an interpretation of the data as a set of elements: points and lines, which considers proximity and continuity to visualize a topological graph. For statistical analysis the L-M or V-D distributed data was represented as boxplots, which boundaries were set at the 25th and 75th percentile. The median is represented as a line in the boxplot. The whisker length is set by the Tukey method, which calculates all the values situated 1.5 times the interquartile distance. Values greater than this range are plotted as individual dots on the graph. Similarly, for determining differences between densities, the distance range from the origin, containing 80% of cumulative MN somas, was determined for every image and represented as boxplots for each $x$ or $y$ distribution.

**Cell culture.** HBMECs (gift of PD Dr Andreas Fischer, DKFZ Heidelberg, Germany) were maintained in 0,1% gelatin-coated flasks in Endopan 3 Kit for endothelial cells (PAN-Biotech, GmbH, Aidenbach, Germany) supplemented with 10% FBS, 100 U ml$^{-1}$ penicillin and 100 μg ml$^{-1}$ streptomycin (both Gibco by Life Technologies, Grand island, NY, USA) in a 5% $CO_2$ humidified incubator at 37 °C. Cells from passages 6 to 10 were used for the experiments.

**Hypoxia-treatment of MN column explants.** MN column explants from E11.5 WT SCs were isolated via open book preparation and placed on growth factor-reduced Matrigel (BD Bioscience) in HBMECs culture medium without VEGF, FGF-2 and FBS (= starvation medium). MN column explants were treated with 150 nM chetomin or solvent (DMSO) for 8 h and then transferred to hypoxic conditions (1% $O_2$) or maintained in normoxic conditions (20% $O_2$) for further 4 h. Subsequently, RNA from the MN explants was isolated for qRT-PCR analysis.

**Conditioned medium (CM).** MN column explants or ingression site (IN) explants from E11.5 WT SC were microdissected via open book preparation and cultured in three-dimensional collagen type I (Corning) in HBMECs culture medium without VEGF, FGF-2 and FBS (= starvation medium). MN-, IN-CM or control medium (without explants) were collected after 48 h and used right away.

**Quantification of human VEGF protein levels by ELISA.** To quantify human VEGF protein levels in NSE-VEGF$^{tg}$ mice, SC from E11.5 NSE-VEGF$^{tg}$ embryos or control littermates were isolated and total cellular protein was extracted using ELISA-extraction buffer (100 mM Tris pH7.5, 150 mM NaCl, 1 mM EDTA, 1% Triton X-100, 0,5% Sodiumdeoxycholate, Protease Inhibitor Cocktail Tablets (Roche Diagnostics)). Subsequently, enzyme-linked immunosorbent assay (ELISA) was performed according to the manufacturer's instructions (R&D Systems, Quantikine DVE00).

**Quantification of mouse VEGF protein levels by ELISA.** Quantification of mouse VEGF protein levels in MN explant and MN-CM was performed according to the manufacturer's instructions (R&D Systems, Quantikine MMV00). Briefly, MN explants of two E11.5 SCs were cultured in 250 μl starvation medium (see above) for 48 h. Total cellular protein of MN explants was extracted using ELISA-extraction buffer. MN-CM (1:5 dilution) was directly subjected to the assay.

**Quantification of mouse Flt1 protein levels by ELISA.** Quantification of mouse Flt1 protein levels was performed according to the manufacturer's instructions (R&D Systems, Quantikine MVR100). Briefly, MN explants of two E11.5 SCs were cultured in 250 μl starvation medium, either in absence or presence of 10 U ml$^{-1}$ heparin (H4784, Sigma-Aldrich) for 20 h. Total cellular protein of MN explants was extracted using ELISA-extraction buffer. MN-CM was directly subjected to the assay.

**Transwell migration assay.** Upon reaching confluency HBMECs were starved for 5 h in HBMECs starvation medium, then trypsinized and plated at $4 \times 10^4$ cells/insert in a Transwell Permeable Support (6.5 mm Insert, 24-well plate, 3.0 μm polycarbonate membrane, tissue culture-treated polystyrene, Costar (Ref: 3415)) with control CM, control CM + 50 ng ml$^{-1}$ VEGF, MN-CM or IN-CM (all with or without adding 1 μg ml$^{-1}$ recombinant mouse Flt1/Fc) in the lower compartment. Inserts were coated before with 10 μg ml$^{-1}$ fibronectin (F0895, Sigma-Aldrich). After 20 h cells were fixed and nuclei were stained. Migration of endothelial cells on the other side of the filter was calculated by counting the number of nuclei per field of view. Five field of views were counted per condition. Quantification was done blind to the experimental condition.

**Tube formation assay.** Tube formation assays were performed in μ-Slide Angiogenesis wells (ibidi GmbH, Germany) using 10 μl of growth factor-reduced Matrigel (BD Bioscience) per well and allowed to polymerize for 30 min at 37 °C. Subsequently, $1 \times 10^4$ HBMECs were resuspended in control CM, control CM + 50 ng ml$^{-1}$ VEGF, MN-CM or IN-CM (all with and without adding 1 μg ml$^{-1}$ recombinant mouse Flt1/Fc), and 50 μl of cell suspension were added per well. Cells were incubated for 4–6 h in a humidified chamber at 37 °C, 5% $CO_2$. Images were acquired using a microscope Zeiss Axiovert 200 M with 5x/0,16 EC Plan-NEOFLUAR Objective. Tube length was quantified using Angiogenesis Analyzer for ImageJ (Gilles Carpentier. ImageJ contribution: angiogenesis analyzer. ImageJ News, 5 October 2012) and number of branches were counted manually. Quantification was done blind to the experimental condition.

**Tube-touching assay.** Tube-touching assays were performed in μ-Slide Angiogenesis wells as described for the tube formation assay except for the addition of either MN explants, IN explants or MN explants from NSE-VEGF$^{tg}$ embryos to cell suspension. 50 μl of HBMECs suspension (prepared as above) including explants were added per well (3–4 explants per well). Cells together with the explants were incubated for 4–6 h in a humidified chamber at 37 °C, 5% $CO_2$. Images were acquired using a microscope Zeiss Axiovert 200 M with 5x/0,16 EC Plan-NEOFLUAR Objective. Numbers of tubes touching the explant were counted manually. Quantification was done blind to the experimental condition.

**Time-lapse video microscopy.** Time-lapse video microscopy was performed for the tube-touching assays. Formation of tubular structures at/around explants were captured every 10 min for an overall duration of 410 min using a Andor/Nikon Ti-TuCam microscope equipped with a Nikon Plan Fluor 10x NA 0.3 objective and a environmental box for controlling $CO_2$ (5%) and temperature (37 °C).

**Scratch assay.** HBMECs ($2,4 \times 10^4$ cells) were seeded into two chambers separated by a divider (Culture-Insert 2 Well in μ-Dish 35 mm, high ibiTreat: ready to use, tissue culture treated, sterilized, ibidi GmbH). For monitoring cell migration, CellTracker Red CMPTX fluorescent dye (Life Technologies) was used. The divider was removed once cells formed a confluent monolayer. MN explants and IN explants were isolated from E11.5 WT SC via open book preparation, analogous for MN explants of NSE-VEGF$^{tg}$ embryos. The explants were placed in the gap. Quantification of the relative gap closure was assessed by measuring the area not covered by HBMECs after 6 h, 20 h and 30 h after initiation of the assay and expressed as percentage of cell coverage of initial scratch area. Quantification was done blind to the experimental condition.

***In silico* analysis.** Cytoscape (an open source bioinformatics software platform, version 3.4.0)[28] with GeneMANIA[63] plugin (version: 3.4.1) was used to generate a list of potential VEGF regulators. VEGF was queried for mouse and human data separately to obtain two lists (one for each species). Co-expression, Co-localization, Genetic interactions, Pathway, Physical interactions, Predicted and Others were chosen as Interaction Networks. The 100-top ranked genes were found using equal weighting for mouse and human data separately. Next, a combined list was generated by merging the genes from both lists. Genes that appear in both lists were counted once. Then they were presented to an already available gene expression data set of SC MNs at E11.5 (ref. 29) (accession identifier: GSE45807). The CEL files were downloaded and processed using statistical language R[64] (version: 3.3.1). To perform background subtraction, quantile normalization and summarization the package *oligo* (version: 1.36.1) was used. To classify expression levels of the genes in question a probability distribution of expression levels for a list of housekeeping genes[65] (Supplementary Data 5) was calculated. The genes with expression level higher than the third or the first quartile were considered as

overexpressed or expressed, respectively. All other genes were classified as underexpressed. To perform an enrichment analysis for stratified gene lists Enrichr[30] was used. The enriched signatures for NCI-Nature 2016 library are presented in the form of a bar graph and clustergram. The same approach as described above was also performed for finding potential regulators of Flt1.

**Statistical analysis.** Statistical analysis was performed using GraphPad Prism (6.0) (GraphPad Software Inc.). All data are expressed as mean ± s.e.m. For comparison between two groups, unpaired Student's $t$-test was used. For comparisons between multiple groups, one-way analysis of variance using Tukey's multiple comparisons test was performed, and $P$ values were adjusted to account for multiple comparisons. For MN distribution statistical analysis Mann–Whitney test was used, due to different variances between groups. Statistical significance was defined as $*P < 0.05$, $**P < 0.01$ $***P < 0.001$, $****P < 0.0001$. The experiments were not randomized. The investigator was blinded for all analysis and quantifications. No samples or animals were excluded from the analysis.

**Data availability.** The authors declare that all data supporting the findings of this study are available within the article and its Supplementary Information Files or from the corresponding author upon reasonable request.

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

## Acknowledgements

We thank Prof. Dr. de Bock and Dr. Lange, Prof. Dr. le Noble, Prof. Dr. Schulte-Merker and Prof. Dr. Mazzone for helpful discussions; PD Dr Fischer for providing HBMECs and Prof. Dr. Carmeliet for providing the plasmids for ISH probes and rVEGF. We also thank the Nikon Imaging Center at the University of Heidelberg and Carl Zeiss Application Centre at DKFZ for providing their imaging facility. We thank Prof. Dr. Pollerberg for her support with the chicken work. C.R.d.A. is supported by ERC (ERC-StG-311367), DFG grant FOR2325 and DFG grant RU 1990/1-1 (SFB/TR23). This work was also supported by DFG grant SFB873. P.H. was supported by a FEBS-Short-Term-Fellowship. Research in the laboratory of E.T.S. is supported by the Swiss National Science Foundation.

## Author contributions

P.H., E.T.S. and C.R.d.A. conceptualized the study. P.H., C.R.d.A. and E.T.S. defined the methodology. Investigation was done by P.H., I.P., A.K., R.L., O.E., E.R. and H.A. Resources were arranged by C.R.d.A., H.H.M and E.T.S. Writing of the original draft was done by P.H. and C.R.d.A. Writing of the review and the editing were done by E.T.S., P.H., H.H.M. and C.R.d.A. Visualization was done by P.H. and C.R.d.A. C.R.d.A. and E.T.S. supervised the study. Project administration was done by C.R.d.A. Funding acquisition was done by C.R.d.A.

## Additional information

**Competing financial interests:** The authors declare no competing financial interests.

