## [Peer Review File · Nature Communications]

Reviewer #1 (Remarks to the Author)

Summary:

The study by Himmels et al., describes the interplay and temporal dynamics between motor neuron (MN)-derived VEGF and sFlt1 in the spinal cord. Using VEGF-overexpressing mice and sFlt1 knockdown strategy in chicken, the group confirms the importance of tight regulation of VEGF for vascular patterning in the context of the spinal cord.

General comment:

The manuscript describes the developmental vascularization of the spinal cord and shows convincingly that VEGF and sFLT1 are involved in blood vessel patterning in this specialized tissue. While the *in vivo* analysis looks solid, and the data are mostly good quality, the manuscript is not strikingly novel. The regulation of VEGF by sFLT1 has been studied extensively in many vascular beds including the CNS, which raises concerns about the conceptual and mechanistic novelty. Studies on the regulation of sFLT1 and VEGF appear mandatory to bridge this mechanistic gap.

Further specific comments:

1. It would be interesting to know why MNs show this type of patterning and whether premature vascularization influences MN function or development.
2. Methodology used throughout the paper could have been more diverse.
3. The authors should define/expand abbreviated terms PNVP, p3, and MN when used for the first time in the text.
4. What does "neuronal domains" mean?
5. Please explain the term "stereotypical blood vessel patterning."
6. The tube touching assay is not very convincing to explain migration, the authors could possibly use other assays such as the Boyden chamber or transwell assays to describe migration of cells towards explants. Otherwise, it would be more insightful to perform the tube touching assay in a time-lapse setting.
7. In this study, microRNAs have been used to reduce expression of FLT1. Although miRNAs can modify gene expression post- transcriptionally, they are not the most efficient tools for gene knockdowns considering they have multiple targets. This raises concerns on the specificity of knockdown approach used here.
8. Line 292, correct spell error "trough" to through.

Reviewer #2 (Remarks to the Author)

In their manuscript, Himmels et al. describe how the initial vasculature of the spinal cord is formed in mouse and chicken embryos. The authors show that motor neurons participate in blood vessel guidance by an autocrine mechanism that regulates the availability of motor neuron-derived VEGF via Flt1 soluble form (sFlt1). The work is important as it identifies the functional role of the spatio-temporal expression of the major angiogenic factor VEGF in the developing spinal cord.

Major issues:

On Fig.1, panels from F to I should show different slices not projections; the yellow color can potentially be misinterpreted as IB4-Nkx2 double positive structures. The pictures on some of these panels are confusing in that it seems that the slices are sampled at different heights along the spinal cord (for example Fig. 2J, as a matter of fact panels I to L do not seem to match with the other panels in this figure). How did the authors decide at which vertical height exactly to perform their analyses? Is the blood vessel patterning - and thus the underlying mechanism dictating this patterning - the same all along the spinal cord? Or are the data and mechanisms presented here specific for a particular region in the spinal cord column?

Fig. 3 A-F: Vessel density at earlier stages should also be shown. Are there any blood vessels at embryonic day 10?

Fig. 3 and Suppl. Fig. S2: The expression of Nse (the promoter of which is used to drive the hVEGF165 in the transgenic mouse) is indeed most pronounced in the MN columns (panels S2 D-F). In contrast, Nse-driven VEGF expression does not show any MN specific expression; in fact, the VEGF expression is all over the spinal cord (panels S2G-H) and casts doubts on the use of this transgenic line to heighten VEGF levels specifically in the MN columns.

As a matter of fact, specific localization of VEGF mRNA (to MNs) is at least hazy throughout the paper: in Fig. 2 E,F,H,I,L there is extensive VEGF mRNA in the tissue immediately surrounding the spinal cord. How can the authors rule out a confounding effect of this VEGF?

Fig. 3 C-H: The increase in blood vessels in the images presented for the NSE-VEGF transgenic seems much more dramatic than the 1.5-fold increase suggested by the quantifications.

Fig. 3 and S5: for the analysis of the angle of vessel ingression it would be helpful to provide representative images of this quantification together with a schematic representation of the respective regions.

Fig. 4A,D: Flt1/Fc treatment under basal conditions already seems to have an effect on tube formation. This is however not reflected by the corresponding quantification.

Fig. 4 A-F: VEGF levels in MN-conditioned medium should be measured and compared to the amounts used in panel B and E. If technically feasible, IN-conditioned medium would serve as an excellent control in this assay.

Fig. 4 N-P: It is not clear if the data for gap closure are corrected for the explant size (bigger explants presumably produce higher amounts of factors and at least panels O and P suggest that there might be a difference in size of the explants). Can other assays - real sprouting assays - be used to further confirm these data? Furthermore, is physical interaction between MN explants and ECs required for the regulation of EC migration? A transwell migration assay would be beneficial to include.

Fig. 5G, This panel is labeled as E11.5 but when comparing to other pictures in the manuscript, it seems to be sampled at an earlier stage. In addition, the NSE ISH should be performed to faithfully discriminate MNs.

Fig. 6, Does mFlt1 and sFlt1 overexpression in MN columns (via the same electroporation technique) result in the opposite effect on vessel ingression and vascularization into and of the spinal cord?

Minor issues:

- 1) Fig. 3, panels A and B should be moved to the supplement, the hemorrhages are not the central finding of this paper and therefore these panels somewhat distract from the main message.
- 2) Fig. 4, panel S should be replaced with a better quality picture.
- 3) The abbreviation PNVP (page 3, line 41) is not introduced.

RESPONSE TO REVIEWER #1

Summary: The study by Himmels et al., describes the interplay and temporal dynamics between motor neuron (MN)-derived VEGF and sFlt1 in the spinal cord. Using VEGF-overexpressing mice and sFlt1 knockdown strategy in chicken, the group confirms the importance of tight regulation of VEGF for vascular patterning in the context of the spinal cord.

General comment: While the *in vivo* analysis looks solid, and the data are mostly good quality, the manuscript is not strikingly novel. The regulation of VEGF by sFLT1 has been studied extensively in many vascular beds including the CNS, which raises concerns about the conceptual and mechanistic novelty. Studies on the regulation of sFLT1 and VEGF appear mandatory to bridge this mechanistic gap.

Answer: Indeed, the regulation of VEGF by sFlt1 has been studied in different vascular beds. However, previous reports have mainly studied sFlt1 expression and function in the endothelium, where sFlt1, expressed in stalk cells of a growing vessel sprout, regulates the pattern of that particular sprout towards an external source of VEGF^{1,2}. The novelty of our findings rely on the special way that motor neurons (MNs) make use of these same molecules to regulate their own vascularisation during spinal cord development. We describe an autocrine mechanism by which MNs express VEGF to allow blood vessel growth, but at the same time express sFlt1 to titrate the availability of their own produced VEGF in order to prevent premature ingression of vessels into motor neuron columns (MN columns). In the new version of our manuscript we have emphasised this novelty better.

To understand the underlying mechanisms of the regulation of sFlt1 and VEGF in MNs we performed an *in silico* analysis in order to identify potential regulators of VEGF, and of Flt1, expressed in MNs during development. For this, we used Cytoscape (an open source software platform to analyse complex networks³) to generate a list of potential regulators/interactors of VEGF and another list for regulators/interactors of Flt1. The obtained lists of genes were then presented to a publicly available dataset of gene expression (mouse genome microarrays) in mouse E11.5 MNs⁴. This allowed us to filter the initial obtained datasets and to generate sub-datasets containing the potential regulators of VEGF or of Flt1 that are expressed in E11.5 spinal cord MNs.

With this generated MN-specific sub-datasets we performed pathway enrichment analysis (using *Enrichr*⁵) to identify the signalling pathways associated with the genes that are expressed and overexpressed in our sub-datasets. This analysis revealed a strong enrichment of known genes involved in VEGF receptor signalling (among them NRP1, see new Supplementary Fig. 9a,b). In addition, enrichment analysis for VEGF indicated that genes involved in the response to oxygen levels/hypoxia (HIF1 α) are highly enriched in MNs at E11.5 (see new Supplementary Fig. 2a,b). Therefore, we focused on these two pathways for further *in vitro* and *in vivo* validation.

a) Regulation of VEGF in motor neurons during embryonic development

To further evaluate the regulation of VEGF, we focused on the potential regulation of *Vegf* expression by HIF1 α . We first determined HIF1 α expression and cellular localisation in MN columns during E10.5-E12.5. We found that HIF1 α is expressed in

MNs at all developmental stages analysed and that its nuclear localisation increases from E10.5 to E11.5, and then decreases again at E12.5 (new Fig. 2a-i).

Consistent with HIF1 α expression and nuclear localisation, certain hypoxia-regulated genes (*Pdk1*, *Egln3* and *Bnip3*) were also expressed in MN columns at those developmental stages (new Supplementary Fig. 2l-n).

We also cultured MN explants in either normoxic (20% O₂) or hypoxic (1% O₂) conditions to show that *Vegf*, as well as *Pdk1*, *Egln3* and *Bnip3*, were increased in MNs cultured in hypoxia (compared to normoxia) conditions (new Fig. 2j-m). Moreover, the increased expression of these genes was HIF1 α -dependent as the presence of a HIF1 α specific inhibitor (chetomin) abolished the mRNA upregulation of those genes (new Fig. 2j-m).

Finally, to show that HIF1 α is responsible for *Vegf* expression in MNs *in vivo*, we injected chetomin into the spinal cord canal of chicken embryos at HH24 (when *Vegf* is expressed in MN columns, floor plate and neuronal progenitors) and analysed *Vegf* expression at HH27. Indeed, *Vegf* expression decreased in MN columns in the presence of the HIF1 α inhibitor (new Fig. 2n,o).

Altogether, these new set of experiments reveal more mechanistic insights and demonstrate that VEGF expression in MNs during spinal cord development, and at the time of its vascularisation, is regulated by HIF1 α .

b) Regulation of sFlt1 in developing motor neurons during embryonic development

Comparison of the list of potential regulators for Flt1 generated by *Cytoscape* with the available gene expression data of E11.5 MNs resulted in a set of genes that, when analysed for enriched pathways, revealed as top hits many of the molecules known to be involved in Flt1 functional regulation (i.e. VEGF, VEGFB, PIGF and NRP1) and signalling in MNs at E11.5 (new Supplementary Fig. 9a,b). We observed that the expression of NRP1 correlated to the expression pattern (new Fig. 7a-k) of Flt1 in MN columns and therefore selected it for further mechanistic insights.

Our analysis showed that NRP1 is highly expressed in MNs at E10.5-E11.5 and its expression decreases, and becomes restricted to certain MN subpopulations, at E12.5 (new Fig. 7a-e). NRP1 expression pattern was similar in chicken (new Fig. 7f-k).

Using two different artificial miRNAs targeting specifically NRP1 in chicken MNs, we could show that NRP1 downregulation results in decreased mRNA expression of *Flt1* (*mFlt1* and *sFlt1*) in MNs, without affecting *Vegf* levels (new Fig. 7l; new Supplementary Fig. 9c). Thus changing the balance between VEGF and sFlt1.

Consistent with a decrease in sFlt1 expression, blood vessel density in MN columns was significantly increased in the artificial NRP1 miRNA electroporated side compared to the non-electroporated one (new Fig. 7m,n; new Supplementary Fig. 9d).

We further performed a rescue experiment to demonstrate the importance of NRP1 regulation of sFlt1. For this we co-electroporated the artificial miRNA for NRP1 together with a plasmid to drive expression of mouse sFlt1 in MNs. Analysis of the vascular phenotype showed no differences between the electroporated and the non-electroporated side, indicating that in the presence of an external source of sFlt1 (non-targetable by the miRNA), downregulation of NRP1 does not lead to premature ingress of blood vessels into MN columns (new Fig. 7o,p).

Collectively, our new data shows that NRP1 transcriptionally controls Flt1 expression, and thus sFlt1 levels.

Further specific comments:

1. It would be interesting to know why MNs show this type of patterning and whether premature vascularization influences MN function or development.

Answer: We have divided this answer in two parts:

Why do MNs show this type of patterning?

During the developmental stages at which we have analysed spinal cord vascularisation several MN developmental processes are taking place. Among them:

- i)* Birth of MNs occurs at the progenitor domain between E9.5-E11.0⁶.
- ii)* MNs segregate into different columnar subtypes that will subsequently innervate different muscles. The segregation into MN subclasses starts at E10.5 and ends at around E12.5⁷.
- iii)* MNs send their axons out of the spinal cord to innervate distinct muscles. Exit of MN axons from the spinal cord starts at E10-10.5 and muscle innervation starts at around E12.5^{8,9}. Formation of neuromuscular junctions occurs between E16.5 and P14¹⁰.
- iv)* MNs do not only make synaptic contact with their muscle partner but they also establish synapses within the spinal cord with interneurons at around E13. Lack of synapses does not mean that there is no neurochemical communication; in fact MNs start showing spontaneous neuronal activity at E11.5-E12.5^{11,12}.

Based on the processes that MNs undergo during the developmental stages analysed, we speculate that MNs show this pattern of vascularisation in order to assure that blood vessels would not interfere in the establishment of the formation of their neural network and their position within. However, once these processes are consolidated and MNs start showing neuronal activity they would allow blood vessel invasion. Indeed, as neuronal activity triggers oxidative metabolism¹³, MNs would require the supply of oxygen and nutrients from blood vessels in order to maintain proper oxidative phosphorylation levels. Thus, by approximately E12 they would allow blood vessel invasion.

Does premature vascularisation influence MN function or development?

We have analysed different aspects of MN development in NSE-VEGF^{tg} embryos and the corresponding WT littermates:

- Analysis of MN counts: Quantification of the number of MNs (stained with Isl1/2 or Foxp1) revealed that MN number does not change in NSE-VEGF^{tg} compared to WT littermates, indicating that birth of MNs, proliferation or cell death are not affected by the premature invasion of blood vessels (new Supplementary Fig. 5e,f).
- Analysis of the MN distribution within MN columns: While in WT embryos different MN subtypes distributed homogenously and in a specific pattern within MN columns, in NSE-VEGF^{tg} MNs showed a more random distribution, with intermixing between different MN subtypes (new Supplementary Fig. 5g-t).
- Analysis of MN axon exit from the spinal cord to the periphery: While MN axons leave the spinal cord in a fasciculated manner, we find that in NSE-VEGF^{tg} embryos this process is disrupted and axons leave the spinal cord highly defasciculated (new Supplementary Fig. 5u-x).

Collectively, while we cannot exclude the possibility that the increased VEGF levels in NSE-VEGF^{tg} might also be part of the observed phenotype, these data suggest that premature presence of blood vessels into MN columns interrupts proper MN distribution and axon projections. Thus, indicating once more that MNs prevent blood vessel ingression during a developmental time window to assure that their developmental processes occur correctly.

2. Methodology used throughout the paper could have been more diverse.

Answer: We believe that we have used the appropriate methodology required for answering our questions. This included the use of two different experimental animal models (mouse and chicken), cell-type-specific gene knock down via *in ovo* electroporation, several *ex vivo* and *in vitro* assays combining tissue explants with

primary endothelial cells in culture, confocal microscopy and image analysis, as well as standard techniques for detecting mRNA and protein expression. In addition, in the revised version of the manuscript we have also included *in silico* bioinformatics analysis, time-lapse microscopy of *in vitro* co-cultures and transwell migration assays.

3. The authors should define/expand abbreviated terms PNVP, p3, and MN when used for the first time in the text.

Answer: We thank the reviewer for pointing this out. In the revised version of our manuscript, the definition of abbreviated terms are now included the first time that the terms are used.

4. What does "neuronal domains" mean?

Answer: We apologise for the confusion that the term "neuronal domains" might have caused. With the term "neuronal domains" we actually meant "neuronal progenitor domains". Indeed, during vertebrate spinal cord development, distinct classes of neurons arise at different positions along the dorsoventral axis, as classified by morphology, function and gene expression pattern⁶. These distinct groups of neurons, localised at different positions along the dorsoventral axis, are termed neuronal progenitor domains. For reasons of clarification we have changed the term "neuronal domains" for "neuronal progenitor domains" in the revised text.

5. Please explain the term "stereotypical blood vessel patterning."

Answer: With "stereotypical blood vessel patterning" we refer to the fact that blood vessels enter and grow inside the developing spinal cord always in a defined manner, such as following a stereotyped pattern. We have further clarified this term in the revised version of the manuscript (see page 3, line 60).

6. The tube touching assay is not very convincing to explain migration, the authors could possibly use other assays such as the Boyden chamber or transwell assays to describe migration of cells towards explants. Otherwise, it would be more insightful to perform the tube touching assay in a time-lapse setting.

Answer: We apologise if there was a misunderstanding regarding the *in vitro* assays used. To understand the behaviour of endothelial cells/blood vessels towards MNs in our original submitted version of our manuscript we had performed three different *in vitro* assays:

1. With the *tube formation assay* we could show that MN-derived conditioned medium was able to induce angiogenesis *in vitro*, and that this was mediated via VEGF secretion (see Fig. 4b-g and Supplementary Fig. 6g-l of our revised manuscript). As suggested by both reviewers, we have now in addition performed *transwell assays* where HBMECs (human brain microvascular endothelial cells) were placed on the upper chamber and subjected to the conditioned medium from MN explants placed in the lower chamber. Quantitative analysis revealed that MN-derived secreted factors are able to induce endothelial cell migration and that this effect is due to VEGF, as the addition of Flt1/Fc to the assay blocks this effect. Thus, similar as with the tube formation assay, transwell experiments also indicate that MNs secrete long-range factors that are pro-angiogenic. As an additional positive control, we also tested the

response of HBMECs to conditioned medium from ingression site (IN) explants and confirmed that HBMECs respond positively to IN secreted factors by increasing their migration and tube formation. These results are now included in Fig. 4a and Supplementary Fig. 6b,i-l of our revised manuscript.

2. To better understand endothelial cell migration in the presence of MNs we had used the modified *scratch assay* (old Figure 4N-T, now Fig. 4k-q), which revealed that in the presence of a MN explant positioned in the center of a scratch, endothelial cells do not migrate and close the scratch as much as in the presence of IN explants. This data indicated that MNs express a membrane-bound or a poorly diffusible factor (potentially associated to the extracellular matrix) that is capable of patterning the vasculature within a short radius of action.
3. Our aim when using the *tube touching assay* was to determine whether endothelial cell tube-like structures (mimicking blood vessels *in vitro*) would be attracted towards a MN explant or repelled and therefore remaining at a distance from the explants. The later possibility would imply that MNs express certain anti-angiogenic factors deposited locally that regulate blood vessel patterning. Our previous results shown in Figure 4 indicated that indeed this is the case. Since, when endothelial cells are co-cultured with MN explants, fewer tube-like structures get close to the explants compared to an explant from a permissive region (IN explant) (old Figure 4K-M, now Fig. 4h-j). As suggested by this reviewer, we have now also performed *time-lapse video microscopy* with this assay. These new results also suggest the presence of locally deposited anti-angiogenic factors in MNs. The supplementary movies provided with the revised manuscript show that while endothelial cells form tubes and these tubes touch and remain close to IN explants (Supplementary Movie 1), in the presence of MN explants endothelial cells still form tubes but fewer of them remain touching the MN explant (Supplementary Movie 2).

7. In this study, microRNAs have been used to reduce expression of FLT1. Although miRNAs can modify gene expression post-transcriptionally, they are not the most efficient tools for gene knockdowns considering they have multiple targets. This raises concerns on the specificity of knockdown approach used here.

Answer:

Spatio-temporal gene knock down in chicken embryos via *in ovo* electroporation can be achieved by electroporation of long double-stranded RNA (dsRNA). Based on the variety of siRNA generated intracellularly this is the most efficient method for gene silencing. However, the use of dsRNA lacks cell-type-specificity and the possibility to label the targeted cells. Thus, we used artificial miRNA constructs driven by the Hb9 promoter to get MN-specific expression of the shRNA-generating construct. The same construct also drives expression of EGFP, and thus, allows for the visualisation of the targeted cells.

Having mentioned the limitations of the approach, we agree with this reviewer that indeed an important issue in the use of miRNA-based plasmids to target specific genes is the specificity.

First, we would like to apologise for not having made clear that the miRNAs used in this study are vector-based *artificial* microRNAs. With the term *artificial* we refer to the fact that we design an artificial 21-22 nucleotide target RNAi sequence towards the gene of

interest. These targeting sequences are designed using the RNAi design software provided by Genscript (http://www.genscript.com/gsfiles/catalog/siRNA_Target_Finding_Strategy.pdf), which assures filtering of up to 95% of off-target effects. With this target-specific RNAi we generate vector-based artificial microRNAs. The methodology used to generate them has been developed by Prof. Esther Stoeckli (a co-author in this manuscript) and published recently¹⁶⁻¹⁹. Briefly, these vector-based artificial miRNAs contain a cell-type-specific promoter (in our case we used the MN-specific Hb9 promoter) to drive the expression of EGFP followed directly by an artificial miR construct derived from miR30 (see ref. 17). The artificial miRNAs are generated by PCR using primers containing miRNA flanking sequences (chicken specific), the 21-22 nucleotides target RNAi sequence and a common loop/stem sequence (from human miRNA30). Subsequently, this PCR product is cloned into the first hairpin site of the Hb9-driven vector (Fig. R2).

Second, to further confirm that the observed phenotype was specifically due to the knock down of Flt1, we have in addition performed the following new experiments:

1. We have further tested a second set of artificial miRNAs for sFlt1 and Flt1 (the latter targeting mFlt1 and sFlt1), which showed similar results as the ones already included in our study. Data generated with this new set of miRNAs are included in the updated Supplementary Fig. 8k-n.
2. In addition, to confirm the specificity of the phenotype we have also performed a rescue experiment where we have co-electroporated a mouse sFlt1-HA expressing plasmid and the chicken artificial sFlt1 miRNA-based plasmid in MNs. In this case, re-expression of sFlt1 rescues the premature vascularisation of MN columns and indicates that the knock down of sFlt1 specifically results in the vascular phenotype observed. This data is also included in the updated Fig. 6i-l.

Finally, our previous results shown in Figure 6 (now Supplementary Fig. 8c-e), using a control construct with a miRNA against the luciferase gene (Hb9-EGFP-miLuc), already showed that the sole harnessing of the cellular miRNA processing pathway due to introduction of a miRNA did not result in a phenotype by itself.

Altogether, we are confident in proposing that the premature ingression of blood vessels into MN columns is due to the downregulation of Flt1 expression and not due to an off-target effect.

8. Line 292, correct spell error "trough" to through.

Answer: The spelling error has been corrected.

References

- 1 Chappell, J. C., Taylor, S. M., Ferrara, N. & Bautch, V. L. Local guidance of emerging vessel sprouts requires soluble Flt-1. *Dev Cell* **17**, 377-386, (2009).
- 2 Zygmunt, T. *et al.* Semaphorin-PlexinD1 signaling limits angiogenic potential via the VEGF decoy receptor sFlt1. *Dev Cell* **21**, 301-314, (2011).
- 3 Cline, M. S. *et al.* Integration of biological networks and gene expression data using Cytoscape. *Nat Protoc* **2**, 2366-2382, (2007).
- 4 Machado, C. B. *et al.* Reconstruction of phrenic neuron identity in embryonic stem cell-derived motor neurons. *Development* **141**, 784-794, (2014).
- 5 Kuleshov, M. V. *et al.* Enrichr: a comprehensive gene set enrichment analysis web server 2016 update. *Nucleic Acids Res* **44**, W90-97, (2016).
- 6 Jessell, T. M. Neuronal specification in the spinal cord: inductive signals and transcriptional codes. *Nat Rev Genet* **1**, 20-29, (2000).
- 7 Francius, C. & Clotman, F. Generating spinal motor neuron diversity: a long quest for neuronal identity. *Cell Mol Life Sci* **71**, 813-829, (2014).
- 8 Ladle, D. R., Pecho-Vrieseling, E. & Arber, S. Assembly of motor circuits in the spinal cord: driven to function by genetic and experience-dependent mechanisms. *Neuron* **56**, 270-283, (2007).
- 9 Lieberam, I., Agalliu, D., Nagasawa, T., Ericson, J. & Jessell, T. M. A Cxcl12-CXCR4 chemokine signaling pathway defines the initial trajectory of mammalian motor axons. *Neuron* **47**, 667-679, (2005).
- 10 Wu, H., Xiong, W. C. & Mei, L. To build a synapse: signaling pathways in neuromuscular junction assembly. *Development* **137**, 1017-1033, (2010).
- 11 Hanson, M. G. & Landmesser, L. T. Characterization of the circuits that generate spontaneous episodes of activity in the early embryonic mouse spinal cord. *J Neurosci* **23**, 587-600, (2003).
- 12 Myers, C. P. *et al.* Cholinergic input is required during embryonic development to mediate proper assembly of spinal locomotor circuits. *Neuron* **46**, 37-49, (2005).
- 13 Kasischke, K. A., Vishwasrao, H. D., Fisher, P. J., Zipfel, W. R. & Webb, W. W. Neural activity triggers neuronal oxidative metabolism followed by astrocytic glycolysis. *Science* **305**, 99-103, (2004).
- 14 Wu, Z. *et al.* Mechanisms controlling mitochondrial biogenesis and respiration through the thermogenic coactivator PGC-1. *Cell* **98**, 115-124, (1999).
- 15 Lin, J., Handschin, C. & Spiegelman, B. M. Metabolic control through the PGC-1 family of transcription coactivators. *Cell Metab* **1**, 361-370, (2005).
- 16 Andermatt, I. *et al.* Semaphorin 6B acts as a receptor in post-crossing commissural axon guidance. *Development* **141**, 3709-3720, (2014).
- 17 Wilson, N. H. & Stoeckli, E. T. Cell type specific, traceable gene silencing for functional gene analysis during vertebrate neural development. *Nucleic Acids Res* **39**, e133, (2011).
- 18 Wilson, N. H. & Stoeckli, E. T. In ovo electroporation of miRNA-based plasmids in the developing neural tube and assessment of phenotypes by Dil injection in open-book preparations. *J Vis Exp*, (2012).
- 19 Wilson, N. H. & Stoeckli, E. T. Sonic hedgehog regulates its own receptor on postcrossing commissural axons in a glypican1-dependent manner. *Neuron* **79**, 478-491, (2013).

RESPONSE TO REVIEWER #2

Summary: In their manuscript, Himmels et al. describe how the initial vasculature of the spinal cord is formed in mouse and chicken embryos. The authors show that motor neurons participate in blood vessel guidance by an autocrine mechanism that regulates the availability of motor neuron-derived VEGF via Flt1 soluble form (sFlt1). The work is important as it identifies the functional role of the spatio-temporal expression of the major angiogenic factor VEGF in the developing spinal cord.

Major issues:

1. On Fig.1, panels from F to I should show different slices not projections; the yellow color can potentially be misinterpreted as IB4-Nkx2.2 double positive structures. The pictures on some of these panels are confusing in that it seems that the slices are sampled at different heights along the spinal cord (for example Fig. 2J, as a matter of fact panels I to L do not seem to match with the other panels in this figure). How did the authors decide at which vertical height exactly to perform their analyses? Is the blood vessel patterning - and thus the underlying mechanism dictating this patterning - the same all along the spinal cord? Or are the data and mechanisms presented here specific for a particular region in the spinal cord column?

Answer: Our aim in Figure 1 was to show how in the spinal cord blood vessel sprouts pattern with respect to distinct neuronal progenitor domains. In order to do so, we imaged 40 µm cryosections so that we could visualise a complete vessel sprout upon the projection of all the confocal slices acquired (information, which would be missed if we just show single confocal slices). Following this reviewer's advice, to clarify that there are no IB4-Nkx2.2 double positive structures and thus no potential misinterpretation, we have now added several single confocal slices for those panels (see new Supplementary Fig. 1q,r,s,t).

Blood vessel patterning is indeed the same along the spinal cord. We now show in Supplementary Fig. 1y-ab the process of spinal cord vascularisation at cervical, brachial, thoracic and lumbar regions of the spinal cord in E11.5 wildtype (WT) mouse embryos. As shown, in all regions blood vessels surround MN columns without invading them.

The analysis of spinal cord vascularisation was always performed at the brachial and thoracic level, taking as a reference the forelimb of the mouse embryo. We decided to analyse blood vessel patterning at these levels, as the vascularisation pattern is similar along the spinal cord, and brachial and thoracic regions are well established and accepted for studying neuronal progenitors, as well as their differentiation and migration in the developing spinal cord^{1,2}.

2. Fig. 3 A-F: Vessel density at earlier stages should also be shown. Are there any blood vessels at embryonic day 10?

Answer: In supplementary Figure 2J-L (now Fig. 3 a-d,i,j), we already showed that at E10.5 there are differences in vessel density when comparing vascularisation in WT and NSE-VEGF^{tg} spinal cords. While in WT embryos no vessels could be seen in MN columns, in NSE-VEGF^{tg} blood vessels had already invaded these areas and blood

vessel density in the spinal cord was significantly increased compared to WT embryos. As requested, to further answer this question, we have now analysed SC vascularisation in WT and NSE-VEGF^{tg} embryos at E9.5. No blood vessels inside the spinal cord could be observed in any of the genotypes (new Supplementary Fig. 4i,j), indicating that at E9.5 the spinal cord is still avascular and that the increased vascularisation in NSE-VEGF^{tg} starts between E9.5 and E10.5. These results are consistent with *Nse* expression, as at E9.5 no signal for *Nse* mRNA can be detected in the spinal cord by ISH (new Supplementary Fig. 4b). We have stated these new results in the revised text (see page 10, line 210-212).

3. Fig. 3 and Suppl. Fig. S2: The expression of *Nse* (the promoter of which is used to drive the *hVEGF165* in the transgenic mouse) is indeed most pronounced in the MN columns (panels S2 D-F). In contrast, *Nse*-driven VEGF expression does not show any MN specific expression; in fact, the VEGF expression is all over the spinal cord (panels S2G-H) and casts doubts on the use of this transgenic line to heighten VEGF levels specifically in the MN columns.

Answer: Unfortunately, the coding sequence of *hVegf*₁₆₅ is highly identical with the *mVegf*₁₆₄ (see Note R1), which prevented us from obtaining a specific ISH probe for *hVegf* without having cross-reaction of *mVegf*. Therefore, as it was indicated in the main text and in the figure legend of old Supplementary Fig. S2 (now Supplementary Fig. 4g,h), the ISH probe used for detecting *Vegf* mRNA does not distinguish between mouse and human *Vegf*, therefore in the new Supplementary Fig. 4g,h the combined expression of both *Vegf* forms is shown (*mVegf* + *hVegf*).

To better demonstrate that there is an increase in VEGF at MN columns and that this increase is hVEGF:

- a) We measured the intensity of the staining in images of *Vegf* ISH from WT and NSE-VEGF^{tg} littermate embryos (treated equally during the entire process of the experiment) at the MN column level (Figure R3a,d,g).
- b) We repeated the *Vegf* ISH in two new sets of embryos and also measured the signal intensity in this new set (Figure R3b,e,h,c,f,i). The obtained results were similar as the previous ones (Figure R3g-i). Similar as before, the ISH shows that *Vegf* (*hVegf* + *mVegf*) mRNA is increased at MN columns and in the ventricular zone of NSE-VEGF^{tg} embryos compared to its expression in WT littermates.

According to the *Nse* expression pattern shown in new Supplementary Fig. 4b-e, the increased expression of *Vegf* at MN column level in NSE-VEGF^{tg} is most probably due to the NSE-driven expression of *hVegf*₁₆₅. The increased expression in the ventricular zone is probably *mVegf* and might be a secondary effect due to the NSE-driven *hVegf* expression in the ventral part of the spinal cord. We have now updated the old Supplementary Figure 2 (now Supplementary Fig. 4g,h) with images of the new ISH and commented this possibility in the figure legend of Supplementary Fig. 4.

- c) *Finally*, to clearly show that *hVegf* is overexpressed in MN columns, we have now microdissected MN columns or explants from the dorsal spinal cord (as internal control where *Nse* is weakly expressed) from WT and NSE-VEGF^{tg} embryos at E11.5 and performed qRT-PCR analysis with primers specific for *hVegf*. qPCR results show that *hVegf* is highly expressed in MN columns of NSE-VEGF^{tg} embryos (new Supplementary Fig. 4f).

Altogether, we are confident that the use of this transgenic line is appropriate to increase VEGF levels in the ventral spinal cord and in particular in MNs.

4. As a matter of fact, specific localization of VEGF mRNA (to MNs) is at least hazy throughout the paper: in Fig. 2 E,F,H,I,L there is extensive VEGF mRNA in the tissue immediately surrounding the spinal cord. How can the authors rule out a confounding effect of this VEGF?

Answer: We apologise for the confusion that we might have caused. Our results do not show that endogenous VEGF is exclusively expressed in MNs but rather that its expression within the spinal cord is restricted to specific areas, being one of them MN columns.

As already reported by others, VEGF is expressed in a wide range of tissues and cellular types in developing mouse embryos. Still, previous reports have shown that VEGF expressed within specific organs is important for the vascularisation of those tissues³⁻⁶.

In our study, (i) the analysis of VEGF expression by ISH, complemented now by qPCR and ELISA in MNs, (ii) the analysis of a CNS-specific transgenic mouse line (NSE-VEGF^{tg}) and (iii) the *in ovo* electroporation approach used to specifically knock down genes in MNs, indicate that MN-derived VEGF is essential for proper blood vessel patterning into the spinal cord. Whether the VEGF derived from tissue outside of the CNS (i.e. the one adjacent to the spinal cord that seems VEGF positive in the old Figure 2E,F,H,I,L (now Fig. 1j-u) might contribute to the observed blood vessel patterning is not known.

5. Fig. 3 C-H: The increase in blood vessels in the images presented for the NSE-VEGF transgenic seems much more dramatic than the 1.5-fold increase suggested by the quantifications.

Answer: We thank the reviewer for pointing it out. We have re-checked the values for blood vessel density obtained for the exact image that was shown in the original Figure 3E-F. Indeed, as this reviewer remarked, while the average of blood vessel density in the entire spinal cord for NSE-VEGF^{tg} embryos is 1.40 ± 0.04 (normalised to WT littermates; n=41), in the chosen image it was 1.63. Also, when considering blood vessel density at MN columns, while the average of NSE-VEGF^{tg} embryos is 1.72 ± 0.08 (normalised to WT littermates; n=41), in the chosen image this value was 1.87. We therefore apologise for this mistake and have selected a better representative image for the quantitative results obtained. To further clarify this issue for this reviewer, we have prepared a figure showing how the vasculature looks in different spinal cord sections from a NSE-VEGF^{tg} embryo (See Figure R4).

6. Fig. 3 and S5: for the analysis of the angle of vessel ingression it would be helpful to provide representative images of this quantification together with a schematic representation of the respective regions.

Answer: As recommended by this reviewer, we have now added to the new Fig. 3 (panels m,n) representative pictures of the procedure used to quantify the angles of vessel ingression, as well as a schematic representation of the respective regions (new Supplementary Fig. 4p).

7. Fig. 4A,D: Flt1/Fc treatment under basal conditions already seems to have an effect on tube formation. This is however not reflected by the corresponding quantification.

Answer: We have re-analysed the image shown in old Figure 4A for control and 4D for Flt1/Fc treatment (new Fig. 4b,c). As shown in Figure R5 quantification of the relative tube length and relative number of branches of this specific image reflects that the values are very similar to the ones shown in the corresponding quantitative graph. Therefore, we have kept those images in our new Fig. 4f,g.

8. Fig. 4 A-F: VEGF levels in MN-conditioned medium should be measured and compared to the amounts used in panel B and E. If technically feasible, IN-conditioned medium would serve as an excellent control in this assay.

Answer: As requested, we have now measured VEGF levels in the resultant conditioned medium via ELISA (MN-conditioned medium generated by culturing MN explants from two E11.5 spinal cords in 250 μ l of medium for 48 h). The results indicate that the VEGF present in MN-CM is at a concentration of $97,96 \pm 3,51$ pg/ml. This concentration is in the range of the VEGF secreted by other types of neurons such as for example Purkinje cells⁷. However, this concentration is much lower than the one used for recombinant VEGF (50 ng/ml) in the migration and transwell assays (new Supplementary Fig. 6b,g,h,k,l). While VEGF concentrations secreted by endogenous tissues might be smaller, it is standard to use concentrations of recombinant VEGF between 10-100 ng/ml to stimulate different cell types in culture.

In addition, as requested by this reviewer we have also performed the tube formation (as well as migration) assay with conditioned medium obtained from ingression site explants. As the new Supplementary Fig. 6b,i-l shows, ingression site conditioned medium is also able to induce migration of HBMECs (Fig. 6b), as well as to induce tube formation in HBMECs (Fig. 6i-l). Interestingly, this effect is not blocked by the addition of Flt1/Fc, suggesting that other pro-angiogenic growth factors might be responsible for this effect. Indeed, Wnt7a/b are expressed at the area where we microdissect the ingression site explants and they have been reported to promote formation and differentiation of the CNS-vasculature⁸. Whether Wnt7a/b are responsible for the observed effect requires further investigation and is out of the scope of the present study. This new data is now discussed in the results section (see page 12-13, line 250-281).

9. Fig. 4 N-P: It is not clear if the data for gap closure are corrected for the explant size (bigger explants presumably produce higher amounts of factors and at least panels O and P suggest that there might be a difference in size of the explants). Can other assays - real sprouting assays - be used to further confirm these data? Furthermore, is physical interaction between MN explants and ECs required for the regulation of EC migration? A transwell migration assay would be beneficial to include.

Answer: The data of the gap closure is not corrected for the explant size, as quantification of the explant size in all the different experimental conditions did not reveal any significant difference. To make this point clearer we have now added a sentence in the revised text stating that the explant size was not different between different experimental conditions. We have also added the graphs with this quantification in the new Supplementary Fig. 6n,o.

With the *tube formation assay* we could show that MN-derived conditioned medium was able to induce angiogenesis *in vitro*, and that this was mediated via VEGF secretion (see Fig. 4 of our revised manuscript). As suggested by reviewer 1 and by this reviewer, we have in addition performed *transwell assays* where HBMECs were placed on the upper chamber and subjected to conditioned medium of MN explants located in the lower chamber. Quantitative analysis revealed that MN-derived factors are able to induce endothelial cell migration and that this effect is due to VEGF, as the addition of Flt1/Fc to the assay blocks this effect. Thus, similar as with the tube formation assay, transwell experiments also indicate that MNs secrete long-range factors that are pro-angiogenic. These results are now included in Fig. 4 and Supplementary Fig. 6.

Altogether, results from those two *in vitro* assays led us to hypothesise that the MN-derived negative factor(s) that regulate blood vessel patterning would need to be locally expressed and that physical or localised interaction was actually needed for proper blood vessel patterning. Indeed, the *modified scratch assay* and *tube touching assay* were designed and performed with the aim of testing this hypothesis. If true, in both assays endothelial cells would remain at a distance or avoid growing towards and touching the explants. Our previous results shown in Figure 4 indeed showed that in the presence of MN explants endothelial cells remain at a distance or do not touch the explant as much as an explant from a permissive spinal cord region (ingression site explant).

As suggested by reviewer 1, to further confirm the data generated with the modified scratch assay and the tube touching assay, we have now performed *time-lapse video microscopy of the tube touching assay*. Analysis of the movies also suggests the presence of locally deposited anti-angiogenic factors in MNs. Indeed, the supplementary movies provided with the revised manuscript show that while endothelial cells form tubes and these tubes touch and remain close to ingression site explants (Supplementary Movie 1), in the presence of MN explants endothelial cells still form tubes but fewer of them remain touching the MN explant (Supplementary Movie 2).

10. Fig. 5G, This panel is labeled as E11.5 but when comparing to other pictures in the manuscript, it seems to be sampled at an earlier stage. In addition, the NSE ISH should be performed to faithfully discriminate MNs.

Answer: As suggested by this reviewer, we have repeated the RNAscope® Multiplex Fluorescent Assay for *mFlt1* and *sFlt1* in new E11.5 embryos to show a similar developmental stage as the other E11.5 analysed in this study. In addition, we have also combined with the RNAscope an immunostaining for MNs to faithfully discriminate them from endothelial cells. As NSE is not a specific marker for MNs, to selectively label MNs we used the anti-Isl1/2 antibody (used throughout this study). The new images showing that indeed *sFlt1* and *mFlt1* are expressed in MNs (Isl1/2⁺ cells), are included in the updated Fig. 5n.

In addition, we have complemented our study by analysing the expression of *mFlt1* and *sFlt1* at E12.5 (when MN columns are invaded by blood vessels). Supporting our working model, while *sFlt1* and *mFlt1* remained expressed in endothelial cells of spinal cord blood vessels, their expression was reduced in MN columns at E12.5. These new results are also included in the updated Fig. 5 (panels o and p).

Finally, we now show evidences demonstrating that sFlt1 is locally retained at MNs vicinity. As some studies have already reported, sFlt1 binds to heparan sulfate proteoglycans anchored to the cell surface or present in the extracellular matrix and can be released by the addition of heparin⁹⁻¹¹. Therefore, we cultured MN explants in the presence or absence of heparin and assessed sFlt1 levels in the conditioned medium by ELISA. Indeed, the addition of heparin to the explants resulted in a significant increase of sFlt1 released to the conditioned medium. Consistently, a reduction of Flt1 present in MN explants after heparin-treatment was also detected. This new data is shown in the new Supplementary Fig. 7c,d.

11. Fig. 6, Does mFlt1 and sFlt1 overexpression in MN columns (via the same electroporation technique) result in the opposite effect on vessel ingression and vascularization into and of the spinal cord?

Answer: To answer this question we generated an Hb9-promoter driven vector to overexpress mouse sFlt1 (HA tagged) in chicken spinal cord MNs via *in ovo* electroporation. Normal blood vessel ingression into MN columns in chicken embryos occurs at HH30. Therefore, if sFlt1 expression remains high in MNs at this developmental stage, we expect that this would prevent the invasion of blood vessels. Indeed, while at HH30 in the control (non-electroporated) spinal cord side blood vessels invaded MN columns, the overexpression of sFlt1-HA in MNs attenuated blood vessel ingression in the electroporated side. These results are now included in the updated Fig. 6m-o.

Minor issues:

1. Fig. 3, panels A and B should be moved to the supplement, the hemorrhages are not the central finding of this paper and therefore these panels somewhat distract from the main message.

Answer: As suggested, those panels have been moved to the supplement (new Supplementary Fig. 4l,m).

2. Fig. 4, panel S should be replaced with a better quality picture.

Answer: The picture shown in panel 4S was taken with equal settings as the one shown in panel 4R. However, due to the limitations of the technique we could technically not avoid some reflection from the needle used to hold the explant in place in this particular explant (Figure R6a,b). To solve this problem we have now repeated the experiment and improved the imaging (Figure R6c-h) by:

- a) Using bright field instead of fluorescence
- b) Imaging the cells with the explant and the needle
- c) Removing the needle and the explant and imaging the cells again right away.
- d) Overlaying both images to be able to mark where the explant was located

The resultant images are now shown in the updated version of the previous figure (new Fig. 4o,p).

3. *The abbreviation PNVP (page 3, line 41) is not introduced.*

Answer: We now properly introduce all abbreviations the first time that they are used.

References

- 1 Balaskas, N. *et al.* Gene regulatory logic for reading the Sonic Hedgehog signaling gradient in the vertebrate neural tube. *Cell* **148**, 273-284, (2012).
- 2 Jessell, T. M. Neuronal specification in the spinal cord: inductive signals and transcriptional codes. *Nat Rev Genet* **1**, 20-29, (2000).
- 3 Hogan, K. A. & Bautch, V. L. Blood vessel patterning at the embryonic midline. *Curr Top Dev Biol* **62**, 55-85, (2004).
- 4 Lammert, E. *et al.* Role of VEGF-A in vascularization of pancreatic islets. *Curr Biol* **13**, 1070-1074, (2003).
- 5 Pierreux, C. E. *et al.* Epithelial: Endothelial cross-talk regulates exocrine differentiation in developing pancreas. *Developmental biology* **347**, 216-227, (2010).
- 6 Raab, S. *et al.* Impaired brain angiogenesis and neuronal apoptosis induced by conditional homozygous inactivation of vascular endothelial growth factor. *Thromb Haemost* **91**, 595-605, (2004).
- 7 Lee, H. *et al.* Pathological roles of the VEGF/SphK pathway in Niemann-Pick type C neurons. *Nature communications* **5**, 5514, (2014).
- 8 Stenman, J. M. *et al.* Canonical Wnt signaling regulates organ-specific assembly and differentiation of CNS vasculature. *Science* **322**, 1247-1250, (2008).
- 9 Orecchia, A. *et al.* Vascular endothelial growth factor receptor-1 is deposited in the extracellular matrix by endothelial cells and is a ligand for the alpha 5 beta 1 integrin. *J Cell Sci* **116**, 3479-3489, (2003).
- 10 Searle, J. *et al.* Heparin strongly induces soluble fms-like tyrosine kinase 1 release in vivo and in vitro--brief report. *Arterioscler Thromb Vasc Biol* **31**, 2972-2974, (2011).
- 11 Sela, S. *et al.* Local retention versus systemic release of soluble VEGF receptor-1 are mediated by heparin-binding and regulated by heparanase. *Circ Res* **108**, 1063-1070, (2011).

Figure R2: Schematic representation of the Hb9-driven artificial miRNA construct used in this study. The plasmid used in this study is driven by a MN-specific promoter (Hb9 promoter) to express a single transcript encoding both a fluorescent protein (enhanced green fluorescent protein (EGFP) to visualise transfected cells) and an artificial miRNA in chicken embryos. The artificial miRNA expression and flanking regions (Flank IV and FlankV) are placed within the context of a miRNA-30-based sequence.

Figure R3: NSE-VEGF^{tg} mouse embryos express higher levels of *Vegf* in the ventral part of the spinal cord. (a-f) Representative images of ISH for *Vegf* (*mVegf* + *hVegf*) in SC sections of WT (a-c) and NSE-VEGF^{tg} (d-f) embryos from three different litters. (Left: images shown in the first submission (litter 1); middle: new litter (litter 2); right: new litter (litter 3) – images shown in the revised Fig. 4g,h). Red circles indicate MN columns. (g-i) Quantification of the ISH intensity at MN columns in the images shown in (a-f), reveal higher levels of *Vegf* in MN columns of NSE-VEGF^{tg} compared to WT embryos. Scale bars 100 μ m.

Figure R4: Spinal cord vascularisation in NSE-VEGF⁹. (a) Mouse embryo scheme showing the region used in this study for analysing SC vascularisation. Sampling heights of SC sections shown in (b-g) are indicated. (b-g) Transverse SC sections at brachial and thoracic level, shown from anterior (b) to posterior (g). Labelled endothelial cells (IB4⁺) at E11.5 for one representative NSE-VEGF⁹ embryo. (h-i) Quantification of blood vessels density in the entire SC (h) or within MN columns (i) for each panel shown in (b-g). Green dotted lines: average of WT embryos analysed in this study; red-dotted lines: average of NSE-VEGF⁹ embryos analysed in this study (see also Fig. 3 in revised manuscript). Scale bars 100 µm.

	Ctrl	Angiogenesis Analyzer	
Ctrl medium			Total length (pixel) 3368
			Rel. tube length 1.197
			Rel. # of branches 1.04
+ Flt1/Fc			Total length (pixel) 3265
			Rel. tube length 1.161
			Rel. # of branches 0.98

Figure R5: Analysis of representative images of tubular-structures. (a,c) Re-analysis of the representative images shown in old Figure 4A,D of tubular-structures formed by HBMECs in either ctrl condition, with (c) or without (a) $1 \mu\text{g ml}^{-1}$ Flt1/Fc. (b,d) Traces/output generated by the imaging analysis software used (Image J using the *Angiogenesis Analyzer* macro) showing the tubes and branches of images in (a) and (c), respectively. Quantitative values are shown in the boxes on the right. Note that values are similar to the ones shown in the corresponding quantitative graph (old Figure 4G,I, now shown in new Fig. 4f,g). Scale bars $100 \mu\text{m}$.

Figure R6: Representative images of modified scratch assay in WT and NSE-VEGF^{tg} embryos. (a-b) Previous representative images shown of scratch assay after 20 h of HBMECs together with MN explant from wildtype (WT) (a) (shown in old Figure 4R) or MN explant from NSE-VEGF^{tg} embryos (b) (image shown in old Figure 4S). e: explant; n: needle; r: reflection. Note that the red spot at the left corner of the image in panel (b) is the reflection of the needle. Explants are artificially overlaid with a coloured shape. (c-h) Representative images of the new experiment performed where we imaged the cells after 20 h using bright field before (c,d) and after removing the needle (e-h). Colour shape placed where explant was located is added to the bright field images (e-h). In (g,h) the dotted line represents the cell front. Images in (g,h) are the representative images shown in new Fig. 4o,p of the revised manuscript. Scale bars 100 μ m.

Motor neurons guide blood vessels in the developing spinal cord via a sFlt1 dependent mechanism

Patricia Himmels, Isidora Paredes, Heike Adler, Andromachi Karakatsani, Robert Luck, Hugo H. Marti, Olga Ermakova, Eugen Rempel, Esther T. Stoeckli and Carmen Ruiz de Almodóvar*

Note R1: mRNA sequence alignment for mouse *Vegfa164* and human *Vegfa165*.

```
#####
# Program: needle
# Rundate: Wed 9 Nov 2016 18:03:47
# Commandline: needle
#   -auto
#   -stdout
#   -asequence emboss_needle-I20161018-124726-0355-31940414-pg.asequence
#   -bsequence emboss_needle-I20161018-124726-0355-31940414-pg.bsequence
#   -datafile EDNAFULL
#   -gapopen 10.0
#   -gapextend 0.5
#   -endopen 10.0
#   -endextend 0.5
#   -aformat3 pair
#   -snucleotidel
#   -snucleotide2
# Align_format: pair
# Report_file: stdout
#####

#=====
#
# Aligned_sequences: 2
# 1: NM_001025250.3 Mus musculus vascular endothelial growth factor A (VEGFA),
# transcript variant 1, mRNA, >gi|160358798:488-1666
# 2: NM_001025366.2 Homo sapiens vascular endothelial growth factor A (VEGFA),
# transcript variant 1, mRNA, >gi|284172447:499-1737
# Matrix: EBLOSUM62
# Gap_penalty: 10.0
# Extend_penalty: 0.5
#
# Length: 1240
# Identity:    1056/1240 (85.2%)
# Similarity:  1056/1240 (85.2%)
# Gaps:       62/1240 ( 5.0%)
# Score: 6374.0#
#
#=====

mouse Vegfa164      1 CTGACGGACAGACAGACAGACACCGCCCCAGCCCCAGCGCCACCTCCT      50
   |||||||||||||||||||||||||||||||||||||||||||||..|||||
human Vegfa165     1 CTGACGGACAGACAGACAGACACCGCCCCAGCCCCAGCTACCACCTCCT      50

mouse Vegfa164     51 CGCCGGCGGGCTGCCGACGGTGGACGCGGCGGCGAGCCGCG---AGGAAC     97
   |.|||||.|||.|||.|||.|||||.|||||.|||||.|||||.|||||.|||..|
human Vegfa165     51 CCCCGCGCGGCGGGACAGTGGACGCGGCGGCGAGCCGCGGGCAGGGGC     100

mouse Vegfa164     98 CGAAGCCCGCGCCCGAGGCGGGGTGGAGGGGTTCGGGGCTCGCGGGATT     147
   ||.|||||||||||||||||||||||||||||||||||||||||||.|||..
human Vegfa165    101 CGGAGCCCGCGCCCGAGGCGGGGTGGAGGGGTTCGGGGCTCGCGGGCTC     150

mouse Vegfa164    148 GCACGGAACCTTTTCGTCCAACCTCTGGGCTCTTCTCGCTCCGTAGTAGC     197
   |||..|..|||||||||||||||||||||||||||||||||.|||.|||.|||
human Vegfa165    151 GCACTGAACCTTTTCGTCCAACCTCTGGGCTGTTCTCGCTTCGGAGGAGC     200

mouse Vegfa164    198 CGTGGTCTGCGCCGAGGAGACAAACCGATCGGAGCTGGGAGAAGTGCTA     247
   |||||..|||.|||.|||.|||.|||.|||.|||.|||.|||.|||.|||..
human Vegfa165    201 CGTGGTCCGCGCGGGGGAAGCCGAGCCGAGCGGAGCCGCGAGAAGTGCTA     250

mouse Vegfa164    248 GCTCGGGCCTGGAGAAGCCGGGGCCCGAGAAGAGAGGGGAG---GAAGAG     294
   |||||||.|||.|||.|||.|||.|||.|||.|||.|||.|||.|||.|||..
human Vegfa165    251 GCTCGGGCAGGAGGAGCCGCGAGCCGAGGAGGGGAGGAGGAAGAAGAG     300

mouse Vegfa164    295 AAGGAAGAGGAGAGGGGGCCGAGTGGGCG-CTCGGCTCTCAGGAGCCGA     343
   |||||||||||||||||||||||||||||||||||||||||.|||.|||.|||..
human Vegfa165    301 AAGGAAGAGGAGAGGGGGCCGAGTGGGCG-CGCGACTCGGCGCTCGGAAGCCG     349
```

mouse Vegfa164	344	GCTCATGGACGGGTGAGGCGGCCGTGTGCGCAGACAGTGTCCAGCCGCG	393
human Vegfa165	350	GCTCATGGACGGGTGAGGCGGCCGTGTGCGCAGACAGTGTCCAGCCGCG	399
mouse Vegfa164	394	CGCGCGCCCCAGGCCCCGGCCGGCCCTCGGTTCCAGAAGGGAGAGGAGC	443
human Vegfa165	400	CGGCTCCCCAGGCCCTGGCCCGGCCCTCGGGCCGGGAGGAAGAGTAGC	449
mouse Vegfa164	444	CCGCCAAGGCGCGCAAGAGAGCGGGCTGCCTCGCAGTCCGAGCCGGAGAG	493
human Vegfa165	450	TCGCCGAGGCGCCGAGGAGAGCGGGCCGCCACAGCCGAGCCGGAGAG	499
mouse Vegfa164	494	GGAGCGGAGCCGCGCCGGCCCGGACGGGCTCCGAAACCATGAACTTT	543
human Vegfa165	500	GGAGCGGAGCCGCGCCGGCCCGGCTCGGGCTCCGAAACCATGAACTTT	549
mouse Vegfa164	544	CTGCTCTCTTGGGTGCACTGGACCTGGCTTTACTGCTGTACCTCCACCA	593
human Vegfa165	550	CTGCTGTCTTGGGTGCATTGGAGCCTTGCCCTGCTGCTCTACCTCCACCA	599
mouse Vegfa164	594	TGCCAAGTGGTCCCAGGCTGCACCCACGACAGA---AGGAGAGCAGAAGT	640
human Vegfa165	600	TGCCAAGTGGTCCCAGGCTGCACCCATGGCAGAAGGAGGAGGGCAGAATC	649
mouse Vegfa164	641	CCCATGAAGTGTCAAGTTTTCATGGATGCTTACCAGCGAAGCTACTGCCGT	690
human Vegfa165	650	ATCACGAAGTGGTGAAGTTTTCATGGATGCTTATCAGCGCAGTACTGCCAT	699
mouse Vegfa164	691	CCGATTGAGACCCTGGTGGACATCTTCCAGGAGTACCCCGACGAGATAGA	740
human Vegfa165	700	CCAATCGAGACCCTGGTGGACATCTTCCAGGAGTACCTGTGATGAGATCGA	749
mouse Vegfa164	741	GTACATCTTCAAGCCCTCCTGTGTGCCGCTGATGCGCTGTGCAGGCTGCT	790
human Vegfa165	750	GTACATCTTCAAGCCATCCTGTGTGCCCTGATGCGATGCGGGGCTGCT	799
mouse Vegfa164	791	GTAACGATGAAGCCCTGGAGTGCCTGCCACGTGAGAGGCAACATCACC	840
human Vegfa165	800	GCAATGACGAGGGCCTGGAGTGTGTGCCACTGAGGAGTCCAACATCACC	849
mouse Vegfa164	841	ATGCAGATCATGCGGATCAAACCTCACCAAAGCCAGCACATAGGAGAGAT	890
human Vegfa165	850	ATGCAGATTATGCGGATCAAACCTCACCAAAGCCAGCACATAGGAGAGAT	899
mouse Vegfa164	891	GAGCTTCTTACAGCACAGCAGATGTGAATGCAGACCAAAGAAAGACAGAA	940
human Vegfa165	900	GAGCTTCTTACAGCACAAATGTGAATGCAGACCAAAGAAAGATAGAG	949
mouse Vegfa164	941	CAAAGCCAGAAAAAATCAGTTCGAGGAAAGGAAAGGGTCAAAAACGA	990
human Vegfa165	950	CAAGACAAGAAAAAATCAGTTCGAGGAAAGGAAAGGGCAAAAACGA	999
mouse Vegfa164	991	AAGCGCAAGAAATCCCGGTTTAAATCCTGGAGCGT-----	1025
human Vegfa165	1000	AAGCGCAAGAAATCCCGGTATAAGTCCCTGGAGCGTGTACGTTGGTGCCCG	1049
mouse Vegfa164	1026	-----TCACTGTGAGCCTT	1039
human Vegfa165	1050	CTGCTGTCTAATGCCCTGGAGCCTCCCTGGCCCCATCCCTGTGGGCTT	1099
mouse Vegfa164	1040	GTTCAGAGCGGAGAAAGCATTTGTTTGTCCAAGATCCGCAGACGTGTAAA	1089
human Vegfa165	1100	GCTCAGAGCGGAGAAAGCATTTGTTTGTACAAGATCCGCAGACGTGTAAA	1149
mouse Vegfa164	1090	TGTTCCCTGCAAAAACAGACTCGCGTTGCAAGGCGAGGCAGCTTGAGTT	1139
human Vegfa165	1150	TGTTCCCTGCAAAAACAGACTCGCGTTGCAAGGCGAGGCAGCTTGAGTT	1199
mouse Vegfa164	1140	AAACGAACGTACTTGCAGATGTGACAAGCCAAGGCGGTGA	1179
human Vegfa165	1200	AAACGAACGTACTTGCAGATGTGACAAGCCAAGGCGGTGA	1239

#-----
#-----

Reviewer #1 (Remarks to the Author)

Himmels and colleagues have submitted a revised version of their previous manuscript, which contains new experiments and explanations. These additions address most of my comments and concerns and provide solid evidence for motor neuron-derived sFlt1 as an important patterning cue during spinal cord vascularization.

Reviewer #2 (Remarks to the Author)

This is a great paper, which certainly deserves publication in top journal like yours. The authors must be congratulated for an excellent revision, which improved the overall impact and novelty of their original study.

Only two minor issues:

1) Unless I missed it (I had problems downloading all files), a graphical abstract summarizing the key findings visually would make the impact of this study even greater.

2) The additional information provided in the last paragraph of comment #1 should be included in the Material and Methods section to clarify for the reader the area of sampling, reference points and rationale for analyzing blood vessel patterning at the brachial and thoracic regions. Along these lines, it may be of benefit to include the data from Fig. R4 (along with a WT embryo for comparison; Supplementary Fig. 1y-ab) to make the point (in the manuscript) that the vascular patterning is comparable along the entirety of the spinal cord analyzed. Alternatively, a comment in the manuscript mentioning that NSE-VEGF₁₂₁-driven blood vessel growth is observed at other levels of the spinal cord would also be sufficient.

RESPONSE TO REVIEWER #1

Summary: *Himmels and colleagues have submitted a revised version of their previous manuscript, which contains new experiments and explanations. These additions address most of my comments and concerns and provide solid evidence for motor neuron-derived sFlt1 as an important patterning cue during spinal cord vascularization.*

Answer: There are no specific comments from reviewer #1.

RESPONSE TO REVIEWER #2

Summary: *This is a great paper, which certainly deserves publication in top journal like yours. The authors must be congratulated for an excellent revision, which improved the overall impact and novelty of their original study.*

Minor comments:

1. Unless I missed it (I had problems downloading all files), a graphical abstract summarizing the key findings visually would make the impact of this study even greater.

Answer: We thank the reviewer #2 for pointing this out. We have checked again all files and made sure that they are not corrupted. We have summarised our key findings in our proposed working model, which serves as a graphical abstract and can be found in Figure 8.

2. The additional information provided in the last paragraph of comment #1 should be included in the Material and Methods section to clarify for the reader the area of sampling, reference points and rationale for analyzing blood vessel patterning at the brachial and thoracic regions. Along these lines, it may be of benefit to include the data from Fig. R4 (along with a WT embryo for comparison; Supplementary Fig. 1y-ab) to make the point (in the manuscript) that the vascular patterning is comparable along the entirety of the spinal cord analyzed. Alternatively, a comment in the manuscript mentioning that NSE-VEGFtg-driven blood vessel growth is observed at other levels of the spinal cord would also be sufficient.

Answer: We thank the reviewer #2 for pointing this out. For further clarification and as recommended, we have now included:

1. the last paragraph of comment #1 in the Methods sections of our revised manuscript (see *Quantification of blood vessel density and ingression angles*).
2. the data from Figure R4 in the new Supplementary Fig. 5h-p.
3. a comment in our revised manuscript stating that the we observed comparable vascularisation defects at different brachial and thoracic levels.